# Smooth Sailing: Lipschitz-Driven Uncertainty Quantification for Spatial Association

**David R. Burt**[*]   **Renato Berlinghieri**[*]   **Stephen Bates**   **Tamara Broderick**
MIT LIDS
{dburt,renb,s_bates,tamarab}@mit.edu

## Abstract

Estimating associations between spatial covariates and responses — rather than merely predicting responses — is central to environmental science, epidemiology, and economics. For instance, public health officials might be interested in whether air pollution has a strictly positive association with a health outcome, and the magnitude of any effect. Standard machine learning methods often provide accurate predictions but offer limited insight into covariate-response relationships. And we show that existing methods for constructing confidence (or credible) intervals for associations can fail to provide nominal coverage in the face of model misspecification and nonrandom locations — despite both being essentially always present in spatial problems. We introduce a method that constructs valid frequentist confidence intervals for associations in spatial settings. Our method requires minimal assumptions beyond a form of spatial smoothness and a homoskedastic Gaussian error assumption. In particular, we do not require model correctness or covariate overlap between training and target locations. Our approach is the first to guarantee nominal coverage in this setting and outperforms existing techniques in both real and simulated experiments. Our confidence intervals are valid in finite samples when the noise of the Gaussian error is known, and we provide an asymptotically consistent estimation procedure for this noise variance when it is unknown.

## 1   Introduction

Scientists and social scientists often seek to understand the direction and magnitude of associations in settings where variables vary spatially. And since these associations are often used to inform policy, communicating uncertainties is crucial. Example associations of interest include the relationship between aerosol concentrations and regional precipitation changes [76], the link between proximity to major highways and the prevalence of dementia [35], and the link between air pollution exposure and birth weight [34]. In each case, the covariates (e.g., aerosol concentrations, proximity to highways, and air pollution) may be viewed as functions of spatial location. Moreover, scientists often have data at some spatial locations but want to infer associations at others. For instance, a country might measure birth weight and air pollution at the municipal level for some municipalities and wish to understand their relationship in municipalities without data.

Our goal in the present work is to provide valid and useful confidence intervals for an estimator of these associations when (i) the underlying model may be misspecified and (ii) inference is needed at spatial locations that may differ from those in the observed data. First we argue that existing methods do not already solve this problem. In particular, we make this argument in turn for modern flexible machine learning methods, linear methods, spatial regression methods, and debiasing approaches.

---

[*]Equal Contribution

39th Conference on Neural Information Processing Systems (NeurIPS 2025).

**Flexible Machine Learning Methods.** A priori, we might expect a nonlinear relationship between air pollution (as a covariate) and birth weight (as the response). A natural idea is to fit a flexible model — such as a (deep) Gaussian process [52, 15], transformer [70, 33], or XGBoost [10]. These methods can achieve high predictive accuracy, but the methods on their own often lack interpretability and do not immediately yield conclusions about covariate-response associations [57, 17]. Researchers have proposed a number of post hoc interpretability methods, such as Shapley values [61, 38], LIME [53], partial dependence plots [20] and Accumulated Local Effects plots [3]. These methods work directly with the fitted model and can be used to describe associations between individual covariates and model predictions. And because they are interpreting the fitted model (rather than the response itself), there is no need (or mechanism) to quantify uncertainties due to insufficient data. When the fitted model closely approximates the true response, one can view the output of these interpretability methods as describing the relationship between the covariates and response.

In many applications (e.g., in engineering, advertising, and marketing), the available data often contains enough information to be confident that the highly flexible machine learning model approximates the response everywhere of interest. Conversely, in many applications in the sciences and social sciences (such as those cited above), there is often insufficient data to confidently fit a nonparametric or high-dimensional model closely to the true (latent) response everywhere relevant. In particular, in many spatial problems, there is often not enough information for a flexible method to reconstruct the response well in many spatial locations of interest. Nonetheless, we might think there could still be sufficient data to capture associations — and, by quantifying uncertainty, we can check our confidence in any conclusions. The discrepancy between applications that are data-rich and data-poor in this sense can help explain the phenomenon observed by Rudin [57]: that post hoc approximations of black-box models may be worse than fitting interpretable models directly. For example, as illustrated in a real-life recidivism case, a linear model used to approximate a black-box model can seriously misrepresent key relationships — including that between race and recidivism.

**Ordinary Least Squares.** So it seems natural to choose an appropriate interpretable model (rather than post hoc method) and quantify its uncertainty. Buja et al. [7] demonstrated that (directly-fit, not post hoc) linear models can be used to interpretably summarize associations even in the face of potentially nonlinear relationships (i.e., under misspecification). This observation helps explain why all of the applied studies cited above use linear regression.[2] Moreover, ordinary least squares (OLS) comes equipped with classical confidence intervals. However, when the linear model is misspecified, Buja et al. [7] notes that the OLS estimator depends on the covariate values, and it follows that classical confidence intervals are valid only at the observed (training) values. Since we're interested in valid intervals at other locations, which will have different covariate values, we need another approach. The sandwich estimator [30, 77, 78, 7] offers valid intervals under misspecification, but only if all covariates are drawn from a single distribution, which isn't the case in our setting where we want to draw inferences at different locations from where we observe data. Thus, even though OLS provides interpretable estimates, it does not solve our key challenge: constructing valid confidence intervals in the common scenario where both (i) the linear model is misspecified and (ii) inference is needed at spatial locations that may differ from those in the observed data.

**Spatial Regression Methods. One might hope that a method specifically tailored for spatial regression could address these issues.** Generalized least squares (GLS) regression [2] is designed to handle spatial correlation in residuals [13, pp. 22–24], but it does not address the bias introduced by misspecification and nonrandom locations. Bayesian spatial models, such as those based on Gaussian processes, are also common, but their credible intervals tend to underestimate uncertainty in the presence of both misspecification and nonrandom locations [74, 42].

**Debiasing Approaches.** Another natural idea is to construct a debiased estimator of the association, and then account for the variance of this debiasing procedure. Importance weighting methods from the covariate shift literature [62] pursue this goal by reweighting each observation according to an estimated density ratio between target and source covariate distributions, aiming to remove bias from distribution shift. Semiparametric inference in partially linear models (e.g., [55, 54, 11]) takes a related approach: it first fits a flexible model to capture variation explained by a "nuisance" parameter (in our case, the spatial location) and then constructs a debiased estimator of the target association by regressing out the variation explained by this nuisance, so that the remaining signal reflects the

---

[2]In fact, Castro Torres and Akbaritabar [9, Figure 1] surveyed quantitative methods in papers up to 2022 and found that over half of those in fields such as agricultural science, social sciences, and health sciences reported results from a linear model.

direct relationship between the covariate and the response. However, these methods face two key limitations. First, they assume that the debiasing step can perfectly remove bias, which need not be true (especially in finite samples). Second, their validity relies on the assumption that observations are drawn independently and identically. And this assumption rarely holds in spatial applications, where locations are fixed and often spread unevenly due to physical, logistical, or policy constraints. We discuss these and related approaches in more detail in Appendix A.

**Our Contribution.** In what follows, we show through real and simulated experiments (Section 4) that existing approaches can yield confidence intervals with coverage far below the nominal level. Our principal contribution is to introduce the first method for constructing confidence intervals in spatial associations that guarantees frequentist coverage at the nominal level even when the underlying model is misspecified and inference is required at fixed, nonrandom locations that differ from those observed. We are able to account for misspecification and nonrandom locations simultaneously by making more spatially appropriate assumptions than prior work. In particular, prior work relies on a (spatially inappropriate) assumption of independent and identically distribution data; we instead assume the response is a smooth function of space observed with homoskedastic Gaussian noise. When the variance of this noise is known, our confidence intervals are valid in finite samples. To address the common case where the variance is unknown, we provide an asymptotically consistent estimator for it. In our experiments, our method is the only one that consistently attains nominal coverage (or even comes close).

## 2   Problem Setup

We start by describing the available data. After reviewing well-specified linear regression, we set up the misspecified case with different target and source data, and we establish our estimand in this case.

**Data.** Following the covariate-shift literature [4, 47, 14], we refer to our fully observed (training) data as the *source* data; likewise, we let *target* data denote the (test) locations and covariates where we do not observe the response but would like to understand the association between covariates and response. In particular, the source data consists of $N$ triplets $(S_n, X_n, Y_n)_{n=1}^N$, with spatial location $S_n \in \mathcal{S}$, covariate $X_n \in \mathbb{R}^P$, and response[3] $Y_n \in \mathbb{R}$. Here $\mathcal{S}$ represents geographic space; we assume $\mathcal{S}$ is a metric space with metric $d_{\mathcal{S}}$. We collect the source covariates in the matrix $X \in \mathbb{R}^{N \times P}$ and the source responses in the $N$-long column vector $Y$. The target data consists of $M$ pairs $(S_m^\star, X_m^\star)_{m=1}^M$, with $S_m^\star \in \mathcal{S}$, $X_m^\star \in \mathbb{R}^P$. The corresponding responses $Y_m^\star \in \mathbb{R}, 1 \le m \le M$ are unobserved. We collect the target covariates in $X^* \in \mathbb{R}^{M \times P}$ and responses in column vector $Y^* \in \mathbb{R}^M$.

**Review: Well-specified Linear Model.** Though we will focus on the misspecified case, we start by reviewing the classic well-specified case for comparison purposes. In the classic well-specified setup, we have $Y_n = \theta_{\text{OLS}}^{\text{T}} X_n + \epsilon_n$ and $Y_m^\star = \theta_{\text{OLS}}^{\text{T}} X_m^\star + \epsilon_m^\star$ with column-vector parameter $\theta_{\text{OLS}} \in \mathbb{R}^P$ and $\epsilon_n, \epsilon_m^\star \overset{\text{iid}}{\sim} \mathcal{N}(0, \sigma^2)$ for some (unknown) $\sigma^2 > 0$. For any fixed set of source data points (and assuming invertibility holds as needed), we can recover the parameter exactly as

$$\theta_{\text{OLS}} = \arg \min_{\theta \in \mathbb{R}^P} \mathbb{E}\Big[ \frac{1}{N} \sum_{n=1}^N (Y_n - \theta^{\text{T}} X_n) \Big] = \mathbb{E}[X^{\text{T}} X]^{-1} \mathbb{E}[X^{\text{T}} Y] = (X^{\text{T}} X)^{-1} X^{\text{T}} \mathbb{E}[Y]. \quad (1)$$

An analogous formula holds for target points; $\theta_{\text{OLS}}$ is constant across covariate values in any case. Since the population expectation is unknown, analysts typically estimate $\theta_{\text{OLS}}$ via maximum likelihood. The standard estimator $(\hat{\theta}_{\text{OLS},p})$ and confidence interval at level $\alpha$ $(I_{\text{OLS},p})$ for the $p$th coefficient $(\theta_{\text{OLS},p})$ are

$$\hat{\theta}_{\text{OLS},p} = e_p^{\text{T}} (X^{\text{T}} X)^{-1} X^{\text{T}} Y, \ I_{\text{OLS},p} = \hat{\theta}_{\text{OLS},p} \pm z_\alpha \rho, \quad (2)$$

where $\rho^2 = \sigma^2 e_p^{\text{T}} (X^{\text{T}} X)^{-1} e_p$, $z_\alpha$ is the $\alpha$-quantile of a standard normal distribution, and $e_p$ denotes the $P$-dimensional vector with a 1 in entry $p$ and 0s elsewhere. Under correct specification, the confidence interval is valid: that is, it provides nominal coverage. E.g., a 95% confidence interval contains the true parameter at least 95% of the time under resampling. The noise variance $\sigma^2$ in $\rho$ used in Eq. (2) is typically replaced with an estimate $\hat{\sigma}^2$.

---

[3]We propose possible extensions to multivariate responses in Appendix C. But we leave the multivariate-response case largely as an area for future work.

**Misspecified Spatial Setup: Data-Generating Process.** In what follows, we assume the data is generated as $Y_n = g(X_n, S_n) + \epsilon_n$ and $Y_m^\star = g(X_m^\star, S_m^\star) + \epsilon_m^\star$, for a function $g$ that need not have a parametric form and with $\epsilon_n, \epsilon_m^\star \overset{\text{iid}}{\sim} \mathcal{N}(0, \sigma^2)$ for some (unknown) $\sigma^2 > 0$.

We assume that source and target covariates are fixed functions of spatial location. Recall that all of the covariates in our examples (aerosol concentrations, proximity to highways, and air pollution) can be expected to vary spatially and be measured with minimal error. We similarly expect meteorological variables such as precipitation, humidity, and temperature to be reasonably captured by this assumption.[4]

**Assumption 1.** *There exists a function $\chi : \mathcal{S} \to \mathbb{R}^P$ such that $X_m^\star = \chi(S_m^\star)$ for $1 \leq m \leq M$ and $X_n = \chi(S_n)$ for $1 \leq n \leq N$.*

Under Assumption 1, our data-generating process simplifies.

**Assumption 2.** *There exists a function $f : \mathcal{S} \to \mathbb{R}$ such that $\forall m \in \{1, \dots, M\}, Y_m^\star = f(S_m^\star) + \epsilon_m^\star$ and $\forall n \in \{1, \dots, N\}, Y_n = f(S_n) + \epsilon_n$, where $\epsilon_m^\star, \epsilon_n \overset{\text{iid}}{\sim} \mathcal{N}(0, \sigma^2)$.*

We emphasize that we are still allowing the response to be a function of both the covariates and the spatial location. But we need not state the dependence on the covariates explicitly in Assumption 2 due to Assumption 1. Formally, let $g$ be the function satisfying $E[Y \mid X, S] = g(X, S)$. By Assumption 1, $X = \chi(S)$, and so $E[Y \mid X, S] = g(\chi(S), S)$. Define $f : \mathcal{S} \to \mathbb{R}$ by $f(s) = g(\chi(S), S)$. In other words, the response can be a function of the covariates, but because the covariates are a fixed function of space, we can also write the response as a function of space alone.

**Our Estimand.** At a high level, our goal is to capture the relationship between a covariate and the response variable at target locations using data from source locations, while taking into account that these two sets of locations may differ. From this perspective, we can define our estimand as the parameter of the best linear approximation to the response, where "best" is defined by minimizing squared error over the target locations.

$$\theta_{\text{OLS}}^\star = \arg\min_{\theta \in \mathbb{R}^P} \mathbb{E}\Big[ \sum_{m=1}^{M} (Y_m^\star - \theta^\top X_m^\star)^2 | S_m^\star \Big]. \tag{3}$$

As in [78, 7], since the data-generating process may be non-linear, $\theta_{\text{OLS}}^\star$ is no longer constant like the well-specified case; instead it is a function of the target locations.

Before solving the minimization in Eq. (3), we argue that covariate shift, as commonly defined, does not solve the problem of interest in this paper. Instead of using our estimand above, one might instead treat the source and target locations as random draws from separate distributions, drop the conditioning on $S_m^\star$ (in Eq. (3)), and rely on covariate-shift methods. Yet to the best of our knowledge these methods currently offer no finite-sample confidence intervals, leaving the question of valid confidence intervals unresolved. Moreover, in the spatial applications we study, locations are rarely i.i.d.: for example, the United States Environmental Protection Agency (EPA) places monitoring stations strategically, and the targets of interest may be only a few municipalities — or even a single one — with missing data. In such settings, it is unclear that a meaningful population-level distribution of locations exists. We therefore condition on the training and target locations in Eq. (3), i.e. we treat the locations as deterministic. Alternatively, we can view our setting as a special case of covariate shift in which the training and target distributions are the respective fixed sets of locations, and therefore have disjoint support. However, under this view, standard covariate-shift estimators are inapplicable.

To solve the minimization in Eq. (3), it will be convenient to assume invertibility, as for OLS.

**Assumption 3.** $X^{\star\text{T}} X^\star$ *is invertible.*

With Assumptions 1 and 3, we can solve the minimization in Eq. (3) to find

$$\theta_{\text{OLS}}^\star = (X^{\star\text{T}} X^\star)^{-1} X^{\star\text{T}} \mathbb{E}[Y^\star | S^\star]. \tag{4}$$

See Appendix F.1 for a derivation. To the best of our knowledge, the target-conditional estimand in Eq. (4) has not been previously proposed or analyzed in spatial linear regression.

---

[4]Conversely demographic covariates may more reasonably be thought of as noisy functions of space, and further work is needed to handle the noisy case.

In order to estimate $\mathbb{E}[Y^\star \mid S^\star]$ sufficiently well to in turn estimate $\theta^\star_{\mathrm{OLS}}$ well, we need to make regularity assumptions. In classical and covariate-shift settings, these take the form of i.i.d. and bounded-density-ratio assumptions. Since we've seen that the classic assumptions are inappropriate in this spatial setting, we instead assume $f$ is shared across source and target (Assumption 2) and not varying so quickly in space as to be difficult to learn from limited data.

**Assumption 4.** *The conditional expectation, $f$, is L-Lipschitz as a function from $(\mathcal{S}, d_\mathcal{S}) \to (\mathbb{R}, |\cdot|)$. That is, for any $s, s' \in \mathcal{S}$, $|f(s) - f(s')| \le L d_\mathcal{S}(s, s')$.*

As an illustration of how this assumption might be satisfied, consider $\chi$ an $L_1$-Lipschitz function of the spatial domain, and $f(s) = \beta^\mathrm{T} \chi(s) + h(s)$ with $h$ a fixed, $L_2$-Lipschitz function of $\mathcal{S}$. Then $f$ is a $(\|\beta\|_2 L_1 + L_2)$-Lipschitz function of the spatial domain. A similar assumption was recently considered in Burt et al. [8] to derive a consistent estimator for prediction error in spatial settings.

Recall (from the discussion of $g$ vs. $f$ after Assumption 2) that we ultimately allow our response to be a function of both the covariates and the spatial location. But, using our assumption that $f$ covers both dependencies (Assumption 2), we impose our smoothness restrictions on the average response as a function of space alone. We think it is often easier to reason about smoothness in physical space, rather than in a (potentially high-dimensional) space of covariates.

## 3 Lipschitz-Driven Inference

We next provide a confidence interval for $\theta^\star_{\mathrm{OLS},p}$, the $p$th coefficient of the target-conditional least squares estimand (Eq. (4)). We support its validity with theory (in the present section) and experiments (in Section 4). To that end, we start by providing an efficiently-computable point estimate. We end by discussing the role and choice of the Lipschitz constant $L$.

**Lipschitz-Driven Point Estimation.** Since the target covariates are known, the key challenge in estimating Eq. (4) is estimating the unknown quantity $\mathbb{E}[Y^\star | S^\star]$.

For our first approximation, recall that, by Assumption 4, $f$ varies smoothly in space. Since the conditional distribution of the responses given the spatial locations is the same function for both target and source data (Assumption 2), we can approximate $\mathbb{E}[Y^\star | S^\star]$ by a weighted average of $\mathbb{E}[Y|S]$ values for locations $S$ near $S^\star$. Concretely, let $\Psi \in \mathbb{R}^{M \times N}$ be a (non-negative) matrix of weights. If $\Psi$ assigns weight mostly to source locations near each corresponding target location, then by the Lipschitz assumption (Assumption 4), $\mathbb{E}[Y^\star | S^\star] \approx \Psi \mathbb{E}[Y|S]$. $\mathbb{E}[Y|S]$ is also unobserved, so we next approximate it by observed values of $Y$ (at each source location in $S$): $\Psi \mathbb{E}[Y|S] \approx \Psi Y$. Together, these two approximations yield the estimator: $\hat{\theta}^\Psi_p = e_p^\mathrm{T} \left(X^{\star\mathrm{T}} X^\star\right)^{-1} X^{\star\mathrm{T}} \Psi Y$. In our experiments, we construct $\Psi$ as follows.

**Definition 5** (Nearest-Neighbor Weight Matrix)**.** *Define the 1-nearest neighbor weight matrix by*

$$\Psi_{mn} = \begin{cases} 1 & S_n = \text{closest source location to } S^\star_m \\ 0 & \text{otherwise} \end{cases} . \text{ We break ties uniformly at random.} \quad (5)$$

While this simple construction works well in our present experiments, we discuss the potential benefits of other constructions in Appendix B.1. This point estimation approach is closely related to KNN imputation [66], used for missing data. But KNN imputation does not account for repeated use of training responses or bias in estimation due to imputation, problems we address in the next section. We provide a more complete discussion of KNN imputation in Appendix A.

**Lipschitz-Driven Confidence Intervals.** We detail how to efficiently compute our proposed confidence interval for $\theta^\star_{\mathrm{OLS},p}$ in Algorithm 1 (Appendix B). We prove its validity in Theorem 7 below. Before stating our result, we establish relevant notation and intuition for how our method works. First, we show the difference between our estimand and the estimator is normally distributed. Toward that goal, we start by writing $\theta^\star_{\mathrm{OLS},p} - \hat{\theta}^\Psi_p = \sum_{m=1}^M w_m f(S^\star_m) - \sum_{n=1}^N v^\Psi_n Y_n$, for $w := e_p^\mathrm{T} \left(X^{\star\mathrm{T}} X^\star\right)^{-1} X^{\star\mathrm{T}} \in \mathbb{R}^M$ and $v^\Psi := w\Psi \in \mathbb{R}^N$. By Assumption 2 and the previous equation,

$$\theta^\star_{\mathrm{OLS},p} - \hat{\theta}^\Psi_p = \underbrace{\sum_{m=1}^M w_m f(S^\star_m) - \sum_{n=1}^N v^\Psi_n f(S_n)}_{\text{bias}} - \underbrace{\sum_{n=1}^N v^\Psi_n \epsilon_n}_{\text{randomness}} . \quad (6)$$

That is, Eq. (6) expresses $\theta^\star_{\text{OLS},p} - \hat{\theta}^\Psi_p$ as the sum of (i) a *bias* term due to differing locations between the source and target data and (ii) a mean-zero Gaussian *randomness* term due to observation noise.

Since the spatial locations are fixed, the bias term is not random and can be written as $b \in \mathbb{R}$. It follows that $\theta^\star_{\text{OLS},p} - \hat{\theta}^\Psi_p \sim \mathcal{N}(b, \sigma^2 \|v^\Psi\|_2^2)$ since the variance of the randomness term is the sum of the variances of its (independent) summands.

Our strategy from here will be to (1) bound $b$, (2) establish a valid confidence interval using our bound on $b$ while assuming fixed $\sigma^2$, and (3) estimate $\sigma^2$ consistently (as $N \to \infty$).

To bound the bias $b$, we use Assumption 4 to write

$$|b| \leq \sup_{g \in \mathcal{F}_L} \left| \sum_{m=1}^M w_m g(S^\star_m) - \sum_{n=1}^N v^\Psi_n g(S_n) \right|, \tag{7}$$

where $\mathcal{F}_L$ is the space of $L$-Lipschitz functions from $\mathcal{S} \to \mathbb{R}$. In Appendix F.3, we show that it is possible to use Kantorovich–Rubinstein duality to restate the right side of Eq. (7) as a Wasserstein-1 distance between discrete measures. This alternative formulation is useful since it can be cast as a linear program [49, Chapter 3]; see Appendix F.3. Let $B$ denote the output of this linear program.

Given $B \geq |b|$, the following lemma (with proof in Appendix F.4) allows us to construct a confidence interval for $\theta^\star_{\text{OLS},p}$ centered on $\hat{\theta}^\Psi_p$. We discuss the benefits of this construction over alternative approaches in Appendix B.5.

**Lemma 6.** *Let $b \in [-B, B]$, $\tilde{c} > 0$, and $\alpha \in (0, 1)$. Then the narrowest $1 - \alpha$ confidence interval that is symmetric and valid for all $\mathcal{N}(b, \tilde{c}^2)$ is of the form $[-B - \tilde{c}\Delta, B + \tilde{c}\Delta]$ where $\Delta$ is the solution of $\Phi(\Delta) - \Phi(-2B/\tilde{c} - \Delta) = 1 - \alpha$ with $\Phi$ the cumulative density function of a standard normal distribution. Also, the $\Delta$ satisfying this inequality is $\Delta \in [\Phi^{-1}(1 - \alpha), \Phi^{-1}(1 - \frac{\alpha}{2})]$.*

The resulting confidence interval appears in Algorithm 1. We next establish its validity. So far, we have covered only the known $\sigma^2$ case. We handle the unknown $\sigma^2$ case after the following theorem.

**Theorem 7.** *Suppose $(S^\star_m, X^\star_m, Y^\star_m)_{m=1}^M$ and $(S_n, X_n, Y_n)_{n=1}^N$ satisfy Assumptions 1 to 4 with known $\sigma^2$. Define the (random) interval $I^\Psi$ as in Algorithm 1 using the known value of $\sigma^2$. Then with probability at least $1 - \alpha$, $\theta^\star_{OLS,p} \in I^\Psi$. That is, $I^\Psi$ has coverage (conditional on the test locations) at least $1 - \alpha$.*

In Appendix F.2, we prove validity of our confidence interval for a generic choice of weight matrix $\Psi$. Theorem 7 is an immediate corollary of that result.

**Consistent Estimation of the Noise Variance $\sigma^2$.** Generally, the noise variance $\sigma^2$ is unknown, so we will substitute an estimate for $\sigma^2$ in the calculation of the confidence interval $I^\Psi$ (Step 5 in Algorithm 1). In Corollary 9 below, we show that the resulting confidence interval has asymptotically valid coverage. To that end, we first show that the estimator in Eq. (10) is consistent for $\sigma^2$.

**Proposition 8.** *Suppose the spatial domain $\mathcal{S} = [-A, A]^D$ for some $A > 0, D \in \mathbb{N}$. Take Assumptions 2 and 4. For any sequence of source spatial locations $(S_n)_{n=1}^\infty$, take $\hat{\sigma}^2_N$ as in Eq. (10). Then $\hat{\sigma}^2_N \to \sigma^2$ in probability as $N \to \infty$.*

See Appendix F.5 for a proof. For intuition, recall that the conditional expectation minimizes expected squared error over all functions. Since the conditional expectation is $L$-Lipschitz (Assumption 4),

$$\sigma^2 = \inf_{g \in \mathcal{F}_L} \mathbb{E}\left[ \frac{1}{N} \sum_{n=1}^N (Y - g(S_n))^2 \Big| S \right]. \tag{8}$$

The empirical version of Eq. (8) is Eq. (10), which we show is a good estimate for $\sigma^2$ for large $N$.

When $\mathcal{S}$ is a subset of Euclidean space or a subset of the sphere, the minimization in Eq. (10) yields a quadratic program for computing $\hat{\sigma}^2_N$. We provide implementation details in Appendix B.7.

Given Proposition 8, it follows from Slutsky's Lemma that the resulting confidence interval is asymptotically valid. We provide a formal statement below, and a proof in Appendix F.5

**Corollary 9.** *For $\mathcal{S}$ as in Proposition 8, with the assumptions and notation of Theorem 7, but with $\sigma^2$ unknown, the confidence interval $I^\Psi$ from Algorithm 1 has asymptotic (with fixed $M$ and as $N \to \infty$) coverage at level $1 - \alpha$.*

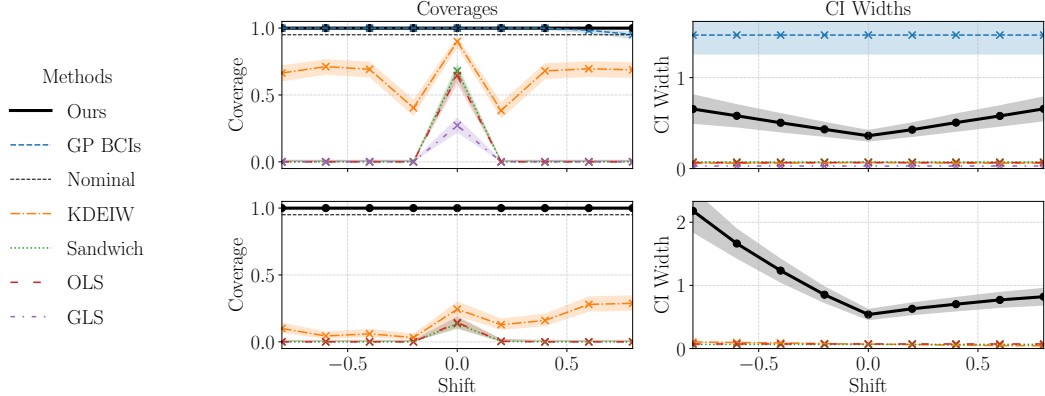

Figure 1: Coverages (left) and confidence interval widths (right) for our method as well as 5 other methods (3 methods in the lower experiment). In the upper experiment, our method and GP BCIs consistently achieve the nominal coverage (95%); the GP BCIs line (dashed blue) overlaps with ours (solid black) for most shifts. Of the two methods with correct coverage, our method yields much narrower intervals. In the lower experiment, only our method achieves the nominal coverage. The shaded region for coverage is a (conservative) 95% confidence interval while the shaded region for CI width is ±2 standard deviations; for more detail, see Appendix D.1.

**Choice of the Lipschitz Constant** $L$. The Lipschitz assumption allows us to make inferences about the target data from the source data. The Lipschitz constant subsequently enters our intervals in two principal ways: via $B$ and via $\hat{\sigma}_N^2$. Intuitively, larger values of $L$ (allowing for less smooth responses) lead to our algorithm constructing confidence intervals with larger bounds ($B$) on the bias but smaller estimated residual variance ($\hat{\sigma}_N^2$). We depict this trade-off in a concrete example in Section 4 (Fig. 2).

Ultimately the choice of Lipschitz constant must be guided by domain knowledge. We give one concrete example describing our choice of the Lipschitz constant in our real-data experiment on tree cover (Section 4). We give a second concrete example of how to select the Lipschitz constant in Appendix B.3; in this case, the response is annual average $PM_{2.5}$ over California. In our simulated experiments, we know the minimum value for which Assumption 4 holds; call it $L_0$. So we first choose $L = L_0$. Then we perform ablation studies in both simulated and real data showing that we essentially maintain coverage while varying $L$ over an order of magnitude around our initial choices. We show that further decreasing $L$ can decrease coverage and discuss why it is useful to err on the side of conservatism (i.e., a larger $L$). However, we expect even small values of $L$ to improve upon classical confidence intervals in terms of coverage, since classical confidence intervals do not account for bias at all; our method similarly ignores bias when $L = 0$.

## 4 Experiments

In simulated and real data experiments, we find that our method consistently achieves nominal coverage, whereas all the alternatives dramatically fail to do so. We also provide ablation studies to evaluate the effect of varying the Lipschitz constant in both simulated and real settings.

**Baselines.** We compare to five alternative constructions.

*Ordinary Least Squares (OLS).* We treat the noise variance as unknown and estimate it as the average squared residual, with a correction for the number of degrees of freedom and a $t$-statistic instead of a $z$-statistic [24, pp. 50–52].

*Sandwich Estimator.* The sandwich estimator [30, 77, 78] uses the same point estimate as OLS but a different variance estimate. We take the variance estimate from MacKinnon and White [39, Equation 6]: $\frac{1}{N-P} e_p^T (X^T X)^{-1} (X^T R X)(X^T X)^{-1} e_p$, where $R$ is a diagonal matrix with the squared residuals as entries.

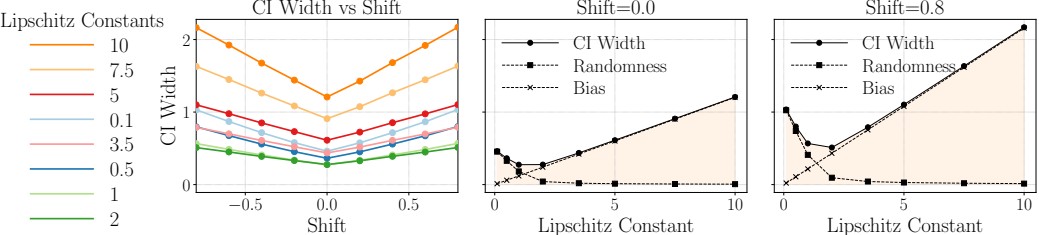

Figure 2: Left: the confidence interval width of our method as a function of shift for each Lipschitz constant $L$. All $L$ yield coverage of $1.0$. Middle and right: the confidence interval width (solid line with dot marker) as a function of the Lipschitz constant for $\text{shift} = 0$ (middle) and $\text{shift} = 0.8$ (right). The vertical axis is shared across all three plots. The bias contribution to the width (dashed line, x marker) is monotonically increasing in $L$. The randomness contribution (dashed line, square marker) is monotonically decreasing.

*Importance Weighted Least Squares (KDEIW).* As suggested by Shimodaira [62, Section 9], we calculate importance weights via kernel density estimation (KDE). We select the bandwidth parameter with 5-fold cross validation; see Appendix B.9. Given the KDE weights, we use the point estimate and confidence interval from weighted least squares.

*Generalized Least Squares (GLS).* We maximize the likelihood of $Y \sim \mathcal{N}(\theta^{\text{T}} X, \Sigma)$, with $\Sigma$ specified by an isotropic Matérn 3/2 covariance function and a nugget, to select the parameters of the covariance function and nugget variance. Then we use the restricted spatial-regression framework [29]; since we project the spatially-correlated error term onto the orthogonal complement of the covariates, the point estimate coincides with OLS.

*Gaussian Process Bayesian Credible Intervals (GP BCIs).* We use the model $Y(S) = \theta^{\text{T}} X(S) + h(S) + \epsilon$, with $\theta^{\text{T}} \sim \mathcal{N}(0, \lambda^2 I_P)$, $h \sim \mathcal{GP}(0, k_\gamma)$, $k_\gamma$ an isotropic Matérn 3/2 kernel function with hyperparameters $\gamma$, and $\epsilon \sim \mathcal{N}(0, \delta^2)$. We select $\{\lambda, \gamma, \delta\}$ by maximum likelihood. We report posterior credible intervals for $\theta_p$.

**Single Covariate Simulation.** In our first simulation, the source locations are uniform on $\mathcal{S} = [-1, 1]^2$ (blue points in Fig. 4 left plots). The target locations are uniform on $[\frac{-1+\text{shift}}{1+|\text{shift}|}, \frac{1+\text{shift}}{1+|\text{shift}|}]$, where shift controls the degree of distribution shift between source and target (orange points in Fig. 4 left plots). In this experiment, the single covariate $X = \chi(S) = S^{(1)} + S^{(2)}$ (Fig. 4, third plot). And the response is $Y = X + \frac{1}{2}((S^{(1)})^2 + (S^{(2)})^2) + \epsilon$, with $\epsilon \sim \mathcal{N}(0, 0.1^2)$. Fig. 4, fourth plot, shows the conditional expectation of the response given location. We can compute the ground truth parameter in closed form because we have access to the conditional expectation of the response (Eq. (4)). We vary $\text{shift} \in [0, \pm 0.2, \pm 0.4, \pm 0.6, \pm 0.8]$ and run 250 seeds for each shift.

Figure 1, top left, shows that only our method and the GP consistently achieve nominal coverage. Given correct coverage, narrower (i.e., more precise) confidence intervals are desirable; Fig. 1, top right, shows that our method yields narrower intervals than the GP. KDEIW comes close to achieving nominal coverage when there is no shift. But under shift with limited data, it is not able to fully debias the estimate, and coverage drops. For large $M$ (here $M = 100$), we expect the sandwich estimator to achieve nominal coverage at $\text{shift} = 0$ since it is guaranteed to cover the population under first-order misspecification without distribution shift [30, 78]. But under any of the depicted non-zero shifts, the sandwich, OLS, and GLS achieve zero coverage. The extreme narrowness of the OLS, sandwich, KDEIW, and GLS intervals (Fig. 1, top right) suggests that the problem with these methods is exactly their overconfidence. Because these approaches assume that the estimator is unbiased (or can be debiased), and that errors are independent and Gaussian, their intervals contract far too quickly, even with small amounts of data (here, $N = 300$). Essentially, these methods' reliance on strong modeling assumptions leads to non-robust coverage.

We note that the confidence intervals and coverages for each method have (approximately) the same values for either $\pm$shift in this experiment (Fig. 1, top right) because the covariate is symmetric around the line $S^{(1)} = S^{(2)}$; see the middle right plot of Fig. 4 in Appendix D. So positive and

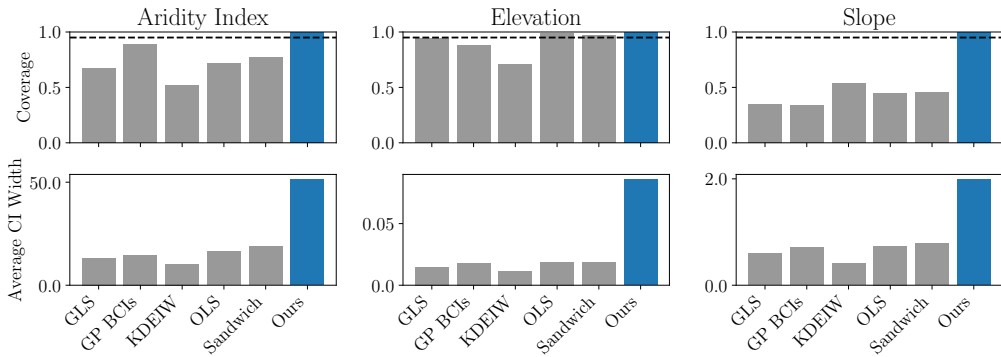

Figure 3: Coverages (upper) and confidence interval widths (lower) for our method as well as 5 other methods. Each column represents a parameter in the tree cover experiment. Only our method consistently achieves the nominal coverage.

negative shift values move the source and target locations symmetrically across the $S^{(1)} = S^{(2)}$ line. We do not expect such a symmetry for general covariates and will not see it in our next simulation (Fig. 1, bottom right).

**Simulation with Several Covariates.** Our second simulation generates locations as in the previous experiment. Now we use $N = 10{,}000$ and $M = 100$. And we generate 3 covariates, $X^{(1)} = \sin(S^{(1)}) + \cos(S^{(2)})$, $X^{(2)} = \cos(S^{(1)}) - \sin(S^{(2)})$ and $X^{(3)} = S^{(1)} + S_2$. The response is $Y = X^{(1)}X^{(2)} + \frac{1}{2}((S^{(1)})^2 + (S^{(2)})^2) + \epsilon$ with $\epsilon \sim \mathcal{N}(0, 0.1^2)$. We focus on inference for the first coefficient. Calculation of $\hat{\sigma}_N^2$ scales poorly with $N$ due to needing to solve a quadratic program. So here we instead estimate $\sigma^2$ using the squared error of leave-one-out 1-nearest neighbor regression fit on the source data. See Appendix B.8 for details. We compare against the same set of methods except we do not include GLS or GP BCIs since these require further approximations to scale for this $N$.

We again find that our method achieves coverage while the other methods do not (Fig. 1, bottom left). In this experiment, no other method achieves coverage over 30% across any shift value (even 0). As before, competing methods are overconfident, with very small CI widths (Fig. 1, bottom right).

For methods besides our own, coverage levels at 0 are generally lower in this experiment at shift $= 0$ than in the previous experiment. The difference is that $N$ is much larger here (making confidence intervals narrower and exacerbating overconfidence), while $M$ is the same. The sandwich estimator covers the analogue of $\theta^\star_{\text{OLS},p}$ where spatial locations are treated as random. So it has good coverage at shift $= 0$ when $M \gg N$, but not when $M \ll N$.

**Effect of Lipschitz Constant on Confidence Intervals in Simulation Experiment.** Above, we know and use the minimum value ($L_0$) for which Assumption 4 holds; that is, $L = L_0$ ($L_0 = 2\sqrt{2}$ and $3\sqrt{2}$, respectively). Now we repeat the first simulation but vary $L \in \{0.1, 0.5, 1.0, 2.0, 3.5, 5, 7.5, 10\}$. All of these $L$ values yield coverage of 1.0 for our method, above the nominal coverage of 0.95, even though coverage is not guaranteed by our theory for $L \leq 2\sqrt{2} \approx 2.8$.

We next show that the confidence interval width reflects a bias-randomness trade-off as $L$ varies. If the noise were known, the confidence interval would be monotonically increasing in $L$. Since the noise is unknown, only the bias contribution to the interval width ($2B$, Step 4 in Algorithm 1) increases (Fig. 2, middle and left, $\times$). Conversely, smaller values of $L$ yield larger values for $\hat{\sigma}_N^2$, so the randomness contribution to the interval width ($2c$, Step 5 in Algorithm 1) increases (Fig. 2, middle and left, $\square$). The full confidence interval width, $2B + 2c\Delta(\alpha)$, is not monotonic in $L$.

**Tree Cover Linear Regression.** We use a linear regression model $Y_{\text{Tree Cover \%}} = \sum_{p \in \mathcal{P}} \theta_p X_p$ to quantify how tree cover percentage in the contiguous United States (CONUS) in the year 2021 relates to three variables, $\mathcal{P} = \{\text{Aridity\_Index}, \text{Elevation}, \text{Slope}\}$. We use the 983 data points from Lu et al.

[37], who in turn draw on [68, 65, 43]. We define our target region in the West portion of CONUS as locations with latitude in the range (25, 50) and longitude in the range (-125, -110). Out of all points in this region, we designate $50\%$ — totaling 133 sites — as target data. Next, we select the source data by taking a uniform random sample of $20\%$ of the remaining spatial locations, repeated over 250 random seeds to assess coverage performance. Each seed yields 170 source locations. Fig. 7 illustrates the spatial split between source and target data for a representative seed. We discuss the data and our pre-processing in detail in Appendix E.

We compare coverage for confidence intervals of the three parameters, $\theta_{\text{Aridity\_Index}}, \theta_{\text{Elevation}}$, and $\theta_{\text{Slope}}$. We discuss how we evaluate coverage in Appendix E.3. In the top row of Fig. 3, we see that our method is the only one to achieve the $95\%$ nominal coverage for all three parameters. Conversely, for the *Slope* parameter, every other method achieves coverage at most $54\%$. In the bottom row of Fig. 10, we again see that alternative methods fail to provide coverage due to their overconfidence (small widths); see also Fig. 10, which shows all methods' constructed confidence intervals across all three parameters for 7 of the 250 seeds. In Appendix B.4, we further discuss how our intervals are the narrowest intervals among those that maintain validity. In Appendix E.5, we conduct a similar analysis but with target locations in the Southeast, rather than West, of CONUS. The results align with our discussion here.

**Choice of Lipschitz Constant in the Tree Cover Experiment.** For the tree cover experiment, we leverage domain knowledge to set the Lipschitz constant to $L = 0.2$, in units of percent tree cover per kilometer (km). This choice implies that a $1\%$ change in tree cover corresponds to moving $1/0.2 = 5$ km. To arrive at this choice, we observe that in certain regions of the U.S., such as the Midwest, tree coverage remains relatively uniform over several kilometers, so smaller Lipschitz constants (e.g., $L = 0.02$, corresponding to a $1\%$ change over 50 km) would be appropriate. However, in other regions — such as the western U.S., where elevation changes are more pronounced (e.g., the Rockies, California, and the Pacific Northwest) — tree cover can change sharply over short distances. To account for these variations conservatively, we choose $L = 0.2$. More generally, for real-world applications, we recommend the following strategy: (i) use domain knowledge to select a reasonable Lipschitz constant for the response variable, and (ii) inflate the Lipschitz constant to ensure a conservative estimate (which is more likely to satisfy Assumption 4).

**Effect of Lipschitz Constant on Confidence Intervals in Tree Cover Experiment.** In Fig. 9, we show the coverage and width of our confidence intervals across three orders of magnitude of Lipschitz constants ($L$ from 0.001 to 1). For $L$ varying between 0.1 and 1 (a single order of magnitude variation around our chosen value of 0.2), we find that coverage is always met except in one case (*Aridity Index* with $L = 0.1$), where it is very close to nominal ($89\%$ instead of $95\%$). For *Slope* and *Elevation*, coverage is met or very nearly met for all $L$ values. For *Aridity Index*, coverage is low for $L \leq 0.1$. Meanwhile, confidence intervals become noticeably wider for $L > 0.5$ while remaining relatively stable for smaller values. These results support our intuition to err on the side of larger $L$ values to be conservative and maintain coverage.

## 5 Discussion

We propose a new method for constructing confidence intervals for spatial linear models. We show via theory and experiments that our intervals accurately quantify uncertainty under misspecification and covariate shift. In experiments, our method is the only method that consistently achieves (or even comes close to) nominal coverage. We observe that, very commonly in spatial data analyses, covariates and responses may be observed at different, nonrandom locations in space. Since our method does not actually use the source covariate values in inference for $\theta_{\text{OLS}}^{\star}$, it can be applied in this common missing-data scenario. Though it requires additional work, we believe the ideas here will extend naturally to the widely used class of generalized linear models.

## Acknowledgments

This work was supported in part by a Social and Ethical Responsibilities of Computing (SERC) seed grant, an Office of Naval Research Early Career Grant, Generali, a Microsoft Trustworthy AI Grant, and NSF grant 2214177.

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

# A Extended Related Work

In this section, we discuss related work on bias in linear regression in spatial settings, local approaches to regression that often rely on data-borrowing strategies similar to our nearest neighbor approach, and covariate shift.

**Bias in Spatial Regression.** Linear models with a Gaussian process random effect — that is, models of the form

$$Y(S) = \theta^{\mathrm{T}} X(S) + g(S) + \epsilon, \tag{9}$$

with $g$ a Gaussian process and $\epsilon$ independent and identically distributed noise — are a classic tool in spatial regression and remain widely used in applications [75, 23, 26]. Whether to treat the covariates as fixed functions of spatial location or random is a topic of significant debate. In order for the model to be identifiable, it is generally necessary for the covariates to be thought of as random [21, Proposition 1]. However, Paciorek [45] observed it may be more reasonable to think of covariates as fixed (the perspective we also take). If $X$ and $g$ are not independent (in the random case) or close to orthogonal in the fixed case, Paciorek [45] illustrates that bias is introduced into the estimation of $\theta$. And that the degree of this bias depends on the relative scales over which $X$ and $g$ vary. He gives a closed form for this bias under strong linear-Gaussian assumptions that depend on parameters that would generally be unknown. Page et al. [46] builds on the result of Paciorek [45], and shows that bias in predictions made by the spatial model due to confounding may be small, despite biases in parameter estimation. Hodges and Reich [29] proposed using restricted spatial regression, essentially explaining as much of the variation in the response as possible by the covariates to reduce bias in estimation due to spatially-correlated error. We take this approach in the GLS baseline used in our experiments. Nobre et al. [44] extended earlier work about bias due to spatial confounding by considering the case when multiple observations are available at the same spatial location with independent noise. They showed that spatial regression models can still be biased with repeated observations due to confounding.

In our work, we focus on bias in estimation of the target conditional OLS. Because we do not assume the model is well-specified, there is no 'true' value of the parameter to estimate. Instead, our goal is to estimate the 'best' (in a least squares sense) linear approximation to the response. We focus on the case when $X$ is a fixed function of location (Assumption 1) and make weak non-parametric, smoothness assumptions (Assumption 4). This contrasts with prior works that generally assume the covariates are random and make linear-Gaussian assumptions to calculate the bias [45, 46]. Our work shows how to directly incorporate bias into confidence intervals for $\theta$ under our weak assumptions, but does not address issues of (non-)identifiability in the estimation problem.

Gilbert et al. [21] showed that despite finite-sample bias in estimation of $\theta$ under spatial confounding, consistent estimation of $\theta$ is possible in the identifiable case when $X$ is random under infill asymptotics. This is essentially orthogonal to our work, as we focus on (finite-sample valid) confidence intervals in the case when $X$ is fixed.

**Local Versus Global Regression Estimates.** An alternative perspective on regression in spatial settings, referred to as *geographically weighted regression* (GWR), focuses on estimating the association between the response and covariates at each location in space [6, 19]. For this regression problem to be well-defined, the covariates at each spatial location must be treated as random variables. Like our method, this approach uses nearest neighbors or local smoothing approaches to estimate the local coefficients at each location in space. And — as in our work — because of this "data-borrowing" from nearby locations, bias is introduced into the estimation of these local regression coefficients. The confidence intervals reported for GWR typically do not account for this bias [79]. While Yu et al. [79] provide a formula for the bias introduced by "data-borrowing", their formula relies on the assumption that the model is well-specified and depends on unobservable parameters.

We focus on the estimation bias in the estimation of global parameters. Although local parameter estimates are often of interest in spatial analyses [19, Chapter 1], global estimates are also useful as summary statistics for making decisions that impact regions or multiple localities. And we characterize the bias in these estimates under weak, non-parametric assumptions and incorporate it into our confidence intervals. Extensions of our approach to GWR coefficients are a promising direction for future research. However, this extension is not simple, as it involves accounting for randomness in the covariates.

**Covariate Shift and Importance Weighting.** Differences between the source locations (with observed responses) and the target locations (at which we want to study the relationship between variables) can be seen as an instance of covariate shift [56, 67, 75]. Covariate shift is often dealt with by importance weighting, reweighting each training example to account for differences in the density of the source and target distributions. Prior work in spatial machine learning has generally focused on addressing covariate shift in the context of estimating a method's prediction error. However, these approaches can, in principle, be extended to parameter estimation in OLS, mirroring the broader covariate shift literature [62].

While in the main text we focused on density ratio estimation using kernel density estimation [62], many other approaches for density ratio estimation can be used [63, 31, 25, 51, 64]. Importance weighting de-biases the estimator, but can only be used if the distribution of the source data is supported on a region containing the test data. And the confidence intervals obtained by importance weighting with estimated weights do not account for errors in estimation of the density ratio (if this exists). These confidence intervals are therefore not guaranteed to achieve the nominal coverage rate. And they are not applicable in cases where extrapolation is required, as the estimator cannot necessarily be de-biased in these cases.

In contrast, our approach ensures nominal coverage even when de-biasing the estimate via importance weighting is not possible. This advantage comes at the cost of an additional regularity assumption (Assumption 4), as well as often empirically wider confidence intervals.

**Comparison to Semiparametric Inference for Partially Linear Model.** Partially linear models take the form

$$\mathbb{E}\left[Y|X,S\right] = \beta^T X + g(S) \mathbb{E}[X \mid S] = \chi(S)$$

where $\beta$ is the parameter of interest, $S$ is a nuisance (or control) variable, $X$ is the covariates, and $g$ is an unknown and possibly complicated function. These models are widely studied in the semiparametric literature. Among many others, Robins et al. [54], Robinson [55], Chernozhukov et al. [11] focus on estimation of $\beta$, under the assumption that the triples $(S_n, X_n, Y_n)$ are independent and identically distributed across data indices $n$.

In many spatial applications, it isn't reasonable to think of the nuisance variable (geographic space) as sampled independently and identically, or even at regularly spaced locations. Observational data are often collected in a highly non-uniform way — densely in some regions, sparsely or not at all in others — due to physical constraints, accessibility, or policy decisions. This non-uniformity introduces distribution shifts when attempting to generalize inferences from one region to another. In the present work, we focus on inference with fixed spatial locations and do not impose regularity conditions on the sampling design. This setup allows us to quantify uncertainty in associations in cases where extrapolation to poorly-sampled geographic areas is required, or in cases with heavily clustered training locations and more uniform target locations.

A notable exception to the assumption of fully i.i.d. data in the semiparametric literature is Heckman [27]. Heckman [27] considered time as a nuisance variable, and assumed this nuisance variable was one-dimensional and sampled densely and in a sufficiently regular way. In contrast, we allow for multiple spatial dimensions and do not require regularity assumptions about the sampling design.

**Comparison to KNN Imputation.** Our estimator is closely related to KNN imputation [66]. The 1-Nearest Neighbor point estimate we consider is equivalent to 1NN imputation for the response, but using only the spatial coordinates — not the covariate values — as features when performing imputation. Directly applying KNN imputation to fill in the target response and then calculating standard confidence intervals would not lead to correct estimates of the variance because of repeated use of training data when imputing missing values — we calculate the variances accounting for potential repeated use of these responses. Additionally, using our smoothness assumptions, we are able to quantify additional uncertainty due to potential bias introduced when imputing the missing response values at the target locations. Because our confidence intervals account for potential bias and calculate the variance accounting for the weight assigned to each training example, our confidence intervals are guaranteed to be conservative, whereas confidence intervals calculated after data imputation need not be.

**Algorithm 1** Lipschitz-Driven CI for $\theta_{\text{OLS},p}^\star$

---

**input** $\{(S_n, X_n, Y_n)\}_{n=1}^N, \{(S_m^\star, X_m^\star)\}_{m=1}^M$, Lipschitz constant $L$, confidence level $1 - \alpha$, $\sigma^2$ (optional)

**output** A $(1 - \alpha)$-confidence interval $I^\Psi$ for $\theta_{\text{OLS},p}^\star$

1:  $\Psi \leftarrow \text{1-NN}(\{S_n\}_{n=1}^N, \{S_m^\star\}_{m=1}^M)$ (Definition 5)
2:  $\hat\theta_p^\Psi \leftarrow e_p^\top \left( X^{\star\top} X^\star \right)^{-1} X^{\star\top} \Psi Y$
3:  $w \leftarrow e_p^\top \left( X^{\star\top} X^\star \right)^{-1} X^{\star\top}, v^\Psi \leftarrow w\,\Psi$
4:  $B \leftarrow \sup_{g \in \mathcal{F}_L} \left| \sum_{m=1}^M w_m\, g(S_m^\star) - \sum_{n=1}^N v_n^\Psi\, g(S_n) \right|$.  Compute with linear program (Appendix B.2).
5:  If $\sigma^2$ unknown, solve for the following estimator via quadratic program (Appendix B.7):
$$\sigma^2 := \hat\sigma_N^2 \;\leftarrow\; \inf_{g \in \mathcal{F}_L} \frac{1}{N} \sum_{n=1}^N \big(Y_n - g(S_n)\big)^2 \tag{10}$$

6:  $c \leftarrow \sigma \, \|v^\Psi\|_2$
7:  Find $\Delta(\alpha)$ satisfying $\Phi\big(\Delta(\alpha)\big) \;-\; \Phi\Big(-2B/c - \Delta(\alpha)\Big) \;=\; 1 - \alpha$, by root finding algorithm (Appendix B.6).
8:  $I^\Psi \;\leftarrow\; \left[ \hat\theta_p^\Psi - B - c\,\Delta(\alpha)\,,\; \hat\theta_p^\Psi + B + c\,\Delta(\alpha) \right]$.

---

# B  Implementation Details

In this section, we describe the implementation details of our method. Particularly, this involves computing the upper bound on the bias, computation of $\Delta$ in Lemma 6 and computation of $\hat\sigma_N^2$ from Eq. (10). To summarize our method, we state it as an algorithm.

## B.1  Selecting the Matrix $\Psi$.

In the main text we selected $\Psi$ by using the nearest source location to each target location. We now provide additional discussion about alternative choices of $\Psi$ and possible trade-offs.

A generalization of the 1-nearest neighbor approach is to consider a $\Psi$ matrix determined by $K$-nearest neighbors.

**Definition 10** (Nearest-Neighbor Weight Matrix). *Define the $K$-nearest neighbor weight matrix by*

$$\Psi_{mn} = \begin{cases} \frac{1}{K} & S_n \in \{K \text{ closest source locations to } S_m^\star\} \\ 0 & \text{otherwise} \end{cases} . \tag{11}$$

*For definiteness, we assume that, if multiple sources are the same distance from a target, ties are broken uniformly at random.*

When using a K-nearest neighbor matrix, there is an inherent bias–variance trade-off in choosing K. Increasing K broadens the geographic range of source observations used, hence the bias will increase; however, it also lowers the variance by averaging over more responses. The degree to which increasing $K$ introduces additional bias depends on the density of data relative to the Lipschitz constant — the benefit of smoother weighting schemes (e.g. $K > 1$) generally becomes more pronounced as the density of spatial sampling increases since smoother weighting schemes naturally leverage multiple nearby points, reducing variance while controlling bias effectively. We do not investigate this trade-off in detail. In the experiments we ran, we found 1-nearest neighbor to work well. This is consistent with results in the mean estimation literature suggesting that if source and target distributions overlap substantially, the variance term remains manageable, even when using 1-nearest neighbor approaches [50].

## B.2  Implementation of the Wasserstein Bound Calculation

We show in Appendix F.3 how Eq. (7) reduces to computing a Wasserstein-1 distance between empirical measures. To implement the Wasserstein distance calculation, we rely on the Python

Optimal Transport library [18]. For the simulated experiments, we compute the cost matrix using Euclidean distances between spatial locations. For the real-world experiment, we use the Haversine distance to account for Earth's curvature.

### B.3 Choice of Lipschitz Constant Based on Prior Research: Example for Air Pollution

As a concrete example of a selecting the Lipschitz constant based on domain-expertise, we consider the problem of selecting a Lipschitz constant in an analysis where the response is annual average $PM_{2.5}$ over California. Chow et al. [12] claim that "Zones of representation for $PM_{2.5}$ varied from 5–10km for the urban Fresno and Bakersfield sites, and increased to 15–20km for the boundary and rural sites" where "[t]he zone of representation is defined as the radius of a circular area in which a species concentration varies by no more than $\pm 20\%$ as it extends outward from the center monitoring site." The annual $PM_{2.5}$ concentrations in the study area do not exceed $30\mu g/m^3$. The combination of a zone of representation between 5–20km, and a variation of not more than 30 $\mu g/m^3$ within this zone of representation suggests a range of possible Lipschitz constants: $0.25$–$1.2(\mu g/m^3)/km$. The authors also point out that topographical and meteorological phenomena contribute to this scale of variation. So we would not expect this proposed constant to be "universal" for problems related to $PM_{2.5}$, but we might expect that this range of Lipschitz constants is a reasonable starting point for other studies involving annual average $PM_{2.5}$ with similar weather and topography to California. We showed in our real-data analysis that a range of Lipschitz constants can still produce qualitatively similar results (Figure 9) and correct coverage. To err towards the side of a conservative analysis, we would recommend a user selects the largest Lipschitz constant in this range (i.e. $1.2(\mu g/m^3)/km$).

### B.4 Confidence-Interval Widths

In some experiments, most notably the tree-cover analysis in Fig. 3, our confidence intervals can appear wide. This increased width arises naturally from our explicit control of bias under extrapolation: when test locations differ substantially from training locations, the Wasserstein-based bias bound enlarges, yielding intervals that faithfully represent genuine uncertainty rather than methodological conservatism. By contrast, existing methods (e.g., OLS, sandwich, and importance-weighted approaches) produce much narrower intervals but fail to achieve nominal coverage, often approaching zero coverage in Fig. 1 and Fig. 2. Our method therefore yields the narrowest intervals among those that do maintain validity. Producing still-narrower intervals would be trivial if one were willing to sacrifice coverage—after all: a zero-width interval achieves minimal size but no inferential meaning. Finally, we note that our intervals are typically informative in practice: in many settings they remain sufficiently tight to exclude zero, providing clear conclusions about the sign and magnitude of associations.

In practice, interval width also reflects the spatial density of available training data. When observations are closely spaced relative to the smoothness scale implied by the Lipschitz constant, the bias bound remains small and the resulting intervals are narrow. As the spacing grows, the bound naturally increases, producing wider intervals that correctly reflect the added uncertainty from extrapolation. Thus, even when data are sparse, the width of our intervals provides a meaningful indication of the reliability of inferences under the assumed smoothness level.

### B.5 Use of Confidence Interval Lemma 6

In all experiments, we construct confidence intervals as in Lemma 6. A simpler alternative, which guarantees $1 - \alpha$ coverage for all Gaussian distributions $\mathcal{N}(b, c^2)$ with $b \in [-B, B]$, is to form two-sided confidence intervals for each $b \in [-B, B]$ and then take their union. This produces an interval of the form

$$[-B \; - \; c\,\Phi^{-1}\big(1 - \tfrac{\alpha}{2}\big) \;\;, \;\; B \; + \; c\,\Phi^{-1}\big(1 - \tfrac{\alpha}{2}\big)]$$

However, the confidence intervals from Lemma 6 are never longer than this union-based approach, and the root-finding step required to compute them adds negligible overhead. Consequently, we opt for the intervals in Lemma 6 in all our experiments.

### B.6 Computation of $\Delta$ for Lemma 6

Define $g(\Delta) = \Phi(\Delta) - \Phi(-\frac{2B}{c} - \Delta) - 1 + \alpha$. Our goal is to find a root of $g$ in the interval $[\Phi^{-1}(1-\alpha), \Phi^{-1}(1-\alpha/2)]$. We first show that $g$ is monotonic, that a root exists in this interval, and that the root is unique.

Differentiating, we see $g'(\Delta) = \phi(\Delta) + \phi(-\frac{2B}{c} - \Delta)$, where $\phi$ denotes the Gaussian probability density function. This is strictly positive, and so $g$ is strictly monotone increasing.

Also,

$$g(\Phi^{-1}(1-\alpha)) = 1 - \alpha - \Phi\left(-\frac{2B}{c} - \Phi^{-1}(1-\alpha)\right) - 1 + \alpha < 1 - \alpha - 1 + \alpha = 0. \quad (12)$$

by non-negativity of the CDF. And,

$$g(\Phi^{-1}(1-\alpha/2)) = 1 - \alpha/2 - \Phi\left(-\frac{2B}{c} - \Phi^{-1}(1-\alpha/2)\right) - 1 + \alpha \quad (13)$$

$$= \alpha/2 - \left(1 - \Phi\left(\frac{2B}{c} + \Phi^{-1}(1-\alpha/2)\right)\right). \quad (14)$$

We used symmetry of the Gaussian in the second equality. Then,

$$\Phi\left(\frac{2B}{c} + \Phi^{-1}(1-\alpha/2)\right) \geq 1 - \alpha/2, \quad (15)$$

and so $g(\Phi^{-1}(1-\alpha/2)) \leq 0$. We conclude that $g$ has a root in the interval. By strict monotonicity of $g$, this root is unique.

We use Brent's method [5] as implemented in Scipy [72] to compute this root numerically.

### B.7 Computation of $\hat{\sigma}_N^2$ via quadratic programming

To estimate the noise parameter, we need to solve the minimization problem

$$\hat{\sigma}_N^2 = \inf_{g \in \mathcal{F}_L} \frac{1}{N} \sum_{n=1}^{N} (Y_n - g(S_n))^2. \quad (16)$$

The first obstacle is that the infimum is taken over an infinite-dimensional space. However, by the Kirszbraun theorem [32, 69], every $L$-Lipschitz function defined on a subset of $\mathbb{R}^D$ or $\mathbb{S}^D$ can be extended to an $L$-Lipschitz function on the whole domain. Since our objective function depends only on the values of $g$ at the source spatial locations, we only need to enforce the Lipschitz condition between all pairs of source spatial locations, not over the entire domain. Enforcing the Lipschitz condition at these $N$ source locations amounts to $N(N-1)/2$ linear inequality constraints.

Particularly, if we define $G = (g(S_1), g(S_2), \ldots g(S_n))$, then we have the constraints

$$AG \leq L\text{vec}(\Gamma) \quad (17)$$

where $L$ is the Lipschitz constant, $\Gamma \in \mathbb{R}^{N^2 - N}$ is the matrix of pairwise distances between distinct points in $S$, and $A \in \mathbb{R}^{(N^2 - N) \times N}$ is a sparse matrix with exactly one 1 and one $-1$ in distinct rows of each column, representing all such pairs in its rows.

The objective is a symmetric, positive-definite quadratic form since it is a sum of squares. Therefore, the optimization problem is a quadratic program.

In practice, we use the Scipy sparse matrix algebra [72] and the CLARABEL solver [22] through the CVXPY optimization interface [16, 1] to solve this quadratic program.

### B.8 Scalable Estimation of $\sigma^2$

The quadratic programming approach for estimating $\sigma^2$ outlined in the main text (Eq. (10)) and described in detail in the previous section does not scale well to large numbers of source locations. Therefore, for our synthetic experiment with $N = 10{,}000$, we take a different approach.

For $1 \leq n \leq N$, let $\eta(n) = \arg\min_{n' \neq n} d_{\mathcal{S}}(S_n, S'_n)$. That is $\eta(n)$ is the index of the nearest point to $S_n$. Define the estimator,

$$\tilde{\sigma}_N^2 = \frac{1}{2N} \sum_{n=1}^{N} (Y_n - Y_{\eta(n)})^2. \tag{18}$$

Then as long as $d_{\mathcal{S}}(S_n, S'_n) \approx 0$, by the Lipschitz assumption, $Y_n - Y_{\eta(n)} \approx \epsilon_n - \epsilon_{\eta(n)}$, and so

$$\tilde{\sigma}_N^2 \approx \frac{1}{2N} \sum_{n=1}^{N} (\epsilon_n - \epsilon_{\eta(n)})^2 = \frac{1}{2N} \left( \sum_{n=1}^{N} \epsilon_n - \sum_{n=1}^{N} \epsilon_n \epsilon_{\eta(n)} + \sum_{n=1}^{N} \epsilon_{\eta(n)}^2 \right). \tag{19}$$

Then,

$$\mathbb{E}[\tilde{\sigma}_N^2] = \sigma^2. \tag{20}$$

In general, we expect the estimate to concentrate around its expectation, provided that no single point in the source data is the nearest neighbor of too many other points in the source data.

### B.9   Implementation of Baseline Methods

We now describe the details of the implementation of the baseline methods.

**Ordinary Least Squares.** We use the ordinary least squares implementation from `statsmodel` [60]. We use the default implementation, which calculates the variance as the average sum of squared residuals with a degrees-of-freedom correction, as in Eq. (2). We use a $t$-statistic to compute the corresponding confidence interval, which is again the default in `statsmodel` [60].

**Sandwich Estimator.** We use the sandwich estimation procedure included in ordinary least squares in `statsmodels` [60]. We use the HC1 function, which implements the sandwich estimator with the degrees-of-freedom correction from Hinkley [28], MacKinnon and White [39]. That is, the variance is estimated as $\frac{1}{N-P} e_p^{\mathrm{T}} (X^{\mathrm{T}} X)^{-1} (X^{\mathrm{T}} R X)(X^{\mathrm{T}} X)^{-1} e_p$. We use the default settings in `statsmodels`, which use a $z$-statistic with the sandwich estimator to compute the corresponding confidence interval.

**Importance Weighted Least Squares.** We use the `scikit-learn` [48] implementation of kernel density estimation to estimate the density of test and train point separately. We use a Gaussian kernel (the default) and perform 5-fold cross validation to select the bandwidth parameter, maximizing the average log likelihood of held-out points in each fold. For the simulation experiments, we performed cross-validation to select the bandwidth parameters over the set $\{0.01, 0.025, 0.05, 0.1, 0.25, 0.5\}$. We selected this set of possible bandwidths to span several orders of magnitude from very short bandwidths, to bandwidths on the same order as the entire domain. We select the bandwidths for the source and target density estimation problems separately. For the real data experiments, the bandwidths was selected from the set $\{0.002, 0.005, 0.01, 0.02, 0.05, 0.1, 0.2, 0.5, 1\}$. This range was selected since the maximum Haversine distance between points in the spatial domain of interest is approximately 1. Once density estimates are obtained, we evaluate the ratio of the density function on the training locations, and use these as weights to perform weighted least squares. Weighted least squares is performed using the default settings in `statsmodels` [60].

**Generalized Least Squares.** We perform generalized least squares in a two-stage manner. We first approximate the covariance structure with a Gaussian process regression model. Then we use this approximation to fit a generalized least squares model with restricted spatial regression [29].

More precisely, first we optimize the maximum likelihood of a Gaussian process regression model with a linear mean function depending on the covariates including an intercept and Matérn 3/2 kernel defined on the spatial source locations. We use the `GPFlow` [40] implementation of the likelihood, and L-BFGS [36] for numerical optimization of the likelihood. The optimization is initialized using the `GPFlow` default parameters for the mean and covariance functions.

Once the maximum likelihood parameters have been found, we use the found prior covariance function for defining the covariance between datapoints to be used in the generalized least squares routine. We use restricted spatial regression [29], and so the covariance matrix is defined as $PKP + \lambda^2 I_N$, where $\lambda^2$ is the noise variance selected by maximum likelihood in the GP model, $K$ is the $N \times N$ matrix formed by evaluating the Matérn 3/2 kernel with the maximum likelihood kernel parameters on the

source locations, and $P$ is the orthogonal projection onto the orthogonal complement of the covariates (including intercept), i.e. $P = I_N - X(X^\mathrm{T}X)^{-1}X^\mathrm{T}$. This covariance matrix $PKP + \lambda^2 I_N$ is passed to the GLS method in statsmodel, and confidence intervals as well as point estimates are computed using the default settings.

**Gaussian Process Bayesian Credible Intervals.**

We first optimize the maximum likelihood of a Gaussian process regression model with Matérn 3/2 kernel defined on the spatial source locations summed with a linear kernel defined on the covariates. This has the same likelihood as having a linear mean function in the covariates with a Gaussian prior over the weights, see Rasmussen and Williams [52, Page 28]. We use the GPFlow [40] implementation of the likelihood, and L-BFGS [36] for numerical optimization of the likelihood. The optimization is initialized using the GPFlow default parameters for the mean and covariance functions. Once we have calculated the maximum likelihood parameters, we compute the posterior credible interval for $\theta$. The posterior over $\theta$ is Gaussian and has the closed form,

$$\theta_\mathrm{post} \sim \mathcal{N}(\Sigma_\mathrm{post}^{-1}X^\mathrm{T}\Sigma^{-1}Y, \Sigma_\mathrm{post}) \tag{21}$$

where

$$\Sigma_\mathrm{post} = \left(X^\mathrm{T}\Sigma^{-1}X + \frac{1}{\lambda^2}I_P\right), \tag{22}$$

where $\lambda^2$ is the prior variance of $\theta$, $\Sigma = K + \delta^2 I_N$, where $\delta^2$ is the variance of the noise, and $K$ is the $N \times N$ kernel matrix formed by evaluating the Matérn 3/2 kernel on the source spatial locations. The posterior for $e_p^\mathrm{T}\theta_\mathrm{post}$ is therefore also Gaussian,

$$e_p^\mathrm{T}\theta_\mathrm{post} \sim \mathcal{N}(e_p^\mathrm{T}\Sigma_\mathrm{post}^{-1}X^\mathrm{T}\Sigma^{-1}Y, e_p^\mathrm{T}\Sigma_\mathrm{post}e_p). \tag{23}$$

Credible intervals are then computed using $z$-scores together with this mean and variance.

## C   Extension To Multivariate Responses

We focus on a scalar response $Y_n \in \mathbb{R}$. We expect similar machinery can be adapted to confidence intervals for each coordinate of $Y_n \in \mathbb{R}^D$. Namely, we can treat each component $Y_n^{(d)} \in \mathbb{R}^D$ as a separate univariate problem, and apply our Lipschitz-based bias bound and variance calibration to each coordinate. If one desires simultaneous coverage over all $D$ outputs, a straightforward Bonferroni correction (i.e. defining $\alpha' = \alpha/D$ and applying our method to construct $1 - \alpha'$ confidence intervals for each coordinate) or another family-wise error control can be used. An exploration of improvements of this Bonferroni correction approach for multivariate responses would be an interesting direction for future work.

## D   Simulations Additional Details

In this section, we present figures to visualize the data generating process for simulation experiments. In all experiments, an intercept is included in the regression as well as the covariates described. All simulation experiments were run on a *Intel(R) Xeon(R) W-2295 CPU @ 3.00GHz* using 36 threads. The total time to run all simulation experiments was under two hours. The single covariate experiment took 9-10 minutes to run; the three covariate experiment took around 80 minutes to run, and the Lipschitz ablation study took around 70 minutes to run.

### D.1   Reported Uncertainty in Simulation Experiments

In Fig. 1, we provide error bars for the empirical coverage as well as a point estimate. The upper side of the confidence interval indicates the largest value of the (true) coverage such that with probability 97.5%, the empirical coverage would be less than or equal to the observed value. Conversely, the lower edge of the confidence interval indicates the smallest value of the (true) coverage such that with probability 97.5%, the empirical coverage would be greater than or equal to the observed value. This is therefore a (conservative) 95% confidence interval for the true coverage.

To calculate the upper and lower bounds, we observe that the empirical coverage is a binomial random variable, with parameters equal to the number of seeds and the true coverage. We then numerically

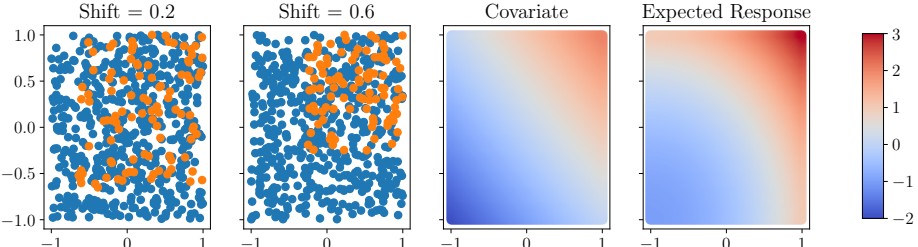

Figure 4: Spatial sites for the source (blue) and target (orange) data are shown in the left most plots for different values of shift used in generating the data. More extreme values of the shift parameter lead to larger biases in parameter estimation from the training data without adjustment. The third plot from the left shows the covariate surface, while the fourth shows the expected response at each spatial location.

invert the binomial cumulative mass function to calculate the upper and lower bounds by performing bisection search on the probability parameter.

We also provide $\pm 2$ standard deviation error bars on the confidence interval width. As the confidence interval width is not necessarily normal this may not be a $95\%$ confidence interval for the confidence interval width. But is meant as an indicator of spread of confidence interval widths each method obtains.

### D.2 Data Generation for the Single Covariate Experiment

We show example datasets used in the simulation experiment with a single covariate and the simulation experiment in which we investigated the impact of Lipschitz constant on confidence interval width and coverage in Fig. 4. The left most two panels show the distribution of source (blue) and target (orange) locations for two values of the shift parameter. Large positive values of the shift parameter lead to target distributions clustered to the top right, values close to zero lead to the target locations being approximately uniformly distributed and large negative values lead to target locations clustered to the bottom left. The third panel from the left in Fig. 4 shows the covariate plotted as a function of spatial location,

$$\chi(S^{(1)}, S^{(2)}) = S^{(1)} + S^{(2)}. \tag{24}$$

The right most panel shows the conditional expectation of the response plotted as a function of spatial location,

$$\mathbb{E}[Y^{\star}|S^{\star} = (S^{(1)}, S^{(2)})] = S^{(1)} + S^{(2)} + \frac{1}{2}((S^{(1)})^2 + (S^{(2)})^2). \tag{25}$$

The gradient of this conditional expectation is,

$$\begin{bmatrix} 1 + S^{(1)} \\ 1 + S^{(2)} \end{bmatrix} \tag{26}$$

which has norm

$$\sqrt{(1 + S^{(1)})^2 + (1 + S^{(2)})^2}. \tag{27}$$

This obtains a maximum of $2\sqrt{2}$ on $[-1, 1]^2$, and so this is the Lipschitz constant of $f(S) = \mathbb{E}[Y|S]$.

### D.3 Data Generation for the Three Covariate Experiment

In Fig. 5 we show data generated for the three covariate shift experiment. Source and target locations are generated as in the one covariate simulation, but with $N = 10{,}000$ ($M = 100$ is still used). These are not shown in Fig. 5.

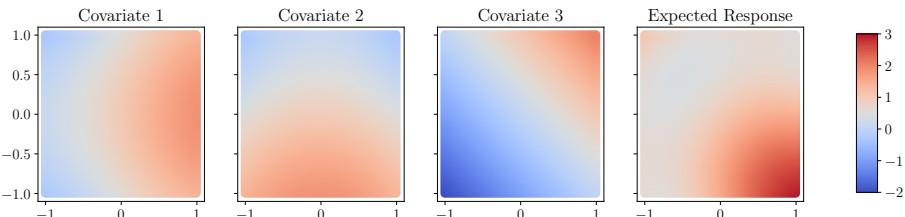

Figure 5: The first 3 plots from the left show the covariate surfaces, while the fourth shows the expected response at each spatial location for the second simulated experiment. The source and target locations (not shown) are the same as in Fig. 4, though with $N = 10,000$.

The covariates are, from left to right in Fig. 5

$$X^{(1)} = \sin(S^{(1)}) + \cos(S^{(2)}) \tag{28}$$

$$X^{(2)} = \cos(S^{(1)}) - \sin(S^{(2)}) \tag{29}$$

$$X^{(3)} = S^{(1)} + S^{(2)}. \tag{30}$$

The conditional expectation of the response (right most panel in Fig. 5) is

$$\mathbb{E}[Y|S = (S^{(1)}, S^{(2)}] = X^{(1)}X^{(2)} + \frac{1}{2}((S^{(1)})^2 + (S^{(2)})^2) \tag{31}$$

$$= (\sin(S^{(1)}) + \cos(S^{(2)}))(\cos(S^{(1)}) - \sin(S^{(2)})) + \frac{1}{2}((S^{(1)})^2 + (S^{(2)})^2). \tag{32}$$

The gradient of this conditional expectation is

$$\begin{bmatrix} \cos(2S^{(1)}) - \sin(S^{(1)} + S^{(2)}) + S^{(1)} \\ -\cos(2S^{(1)}) + \sin(S^{(1)} - S^{(2)}) + S^{(2)} \end{bmatrix}. \tag{33}$$

We see that both arguments of this gradient are less than 3 in absolute value, and therefore the norm of this Lipschitz constant is less than or equal to $\sqrt{3^2 + 3^2} = 3\sqrt{2}$.

# E  Tree Cover Experiment Additional Details

In this section we provide additional details for the real data tree cover experiment, as well as figures to visualize the data and additional experimental results. The tree cover experiments was run on a *Intel(R) Xeon(R) W-2295 CPU @ 3.00GHz* using 36 threads. The total time to run the experiment was under 5 minutes. The Lipschitz ablation study also took less than 5 minutes to run.

## E.1  Tree Cover Data

Our analysis draws on data from Lu et al. [37], who manually labeled 983 high-resolution Google Maps satellite images for tree cover percentage. While no license is specified, we used this data with permission of the authors. These images were selected from random locations within the 2021 USFS Tree Canopy Cover (TCC) product [68]. This dataset is public domain. We follow Lu et al. [37] and use three covariates:

1. *Global Aridity Index (1970–2000)*: Averaged at a 30 arc-seconds resolution [65]. This index is calculated as the ratio of precipitation to evapotranspiration, with lower values indicating more arid conditions.

2. *Elevation*: Provided by NASA's 30-meter resolution dataset [43].

3. *Slope*: NASA's 30 m Digital Elevation Model. Also provided by NASA's 30-meter resolution dataset [43]. While we could not find specific license information, as the *Slope* and *Elevation* datasets are produced by a US government agency (NASA), we understand this data to be public domain following section 105 of the Copyright Act of 1976.

Figure 6 provides a visual overview of both the tree cover and covariates. As a preprocessing step, we convert the (latitude, longitude) coordinates of each data point into radians. This allows us to use the Haversine formula to compute distances in kilometers for the Wasserstein-1 cost and the nearest neighbor weighting procedure.

Elevation and slope are important factors influencing tree cover worldwide. Generally, areas at lower elevations tend to have more tree cover, often due to higher temperatures [41], and sloped terrains also support greater tree coverage [59]. As done in Lu et al. [37], we focus on these three covariates and did not include additional factors that might affect tree cover. This decision was made because our primary objective is to demonstrate uncertainty quantification rather than to provide a comprehensive explanation of tree cover dynamics.

Figure 6 provides a visual overview of the distribution for both the tree cover and covariates.

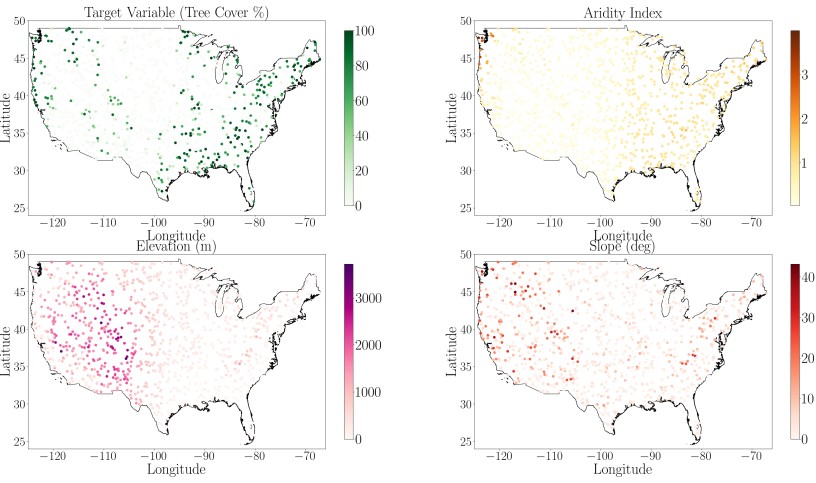

Figure 6: Tree cover response and covariates. The dots represent the 983 locations considered. Top left: distribution of tree cover percentage. Top right: Average Aridity Index, measured as the ratio of precipitation to evapotranspiration. Bottom left: Elevation, measured in meters. Bottom right: Slope, measured in degrees.

### E.2 Source and Target Data Split and Spatial Preprocessing

We define our target region to be the Western portion of the Continental United States. In particular, we consider locations that have latitudes between $25°$ and $50°$ and longitudes between $-125°$ and $-110°$. Within spatial locations in this defined region, we pick $50\%$ of all spatial locations — totaling 133 sites — as *target* data.

To select the *source* data, we perform the following steps:

1. Consider all the remaining spatial locations, i.e. exclude the 133 target points from the pool of 983 spatial points. This leaves us with 850 points.

2. Since we are interested in evaluating whether our method and the baselines achieve nominal coverage, uniformly randomly sample $20\%$ of the remaining locations across 250 different random seeds. By doing this, for each random seed we have 170 source locations.

Fig. 7 visually represents the spatial distribution of the source and target locations for one representative random seed.

As a preprocessing step, we also convert the geographical coordinates (latitude and longitude) of each data point from degrees to radians. This conversion is essential because it allows us to apply the Haversine formula, which calculates the great-circle distance between two points on the Earth's

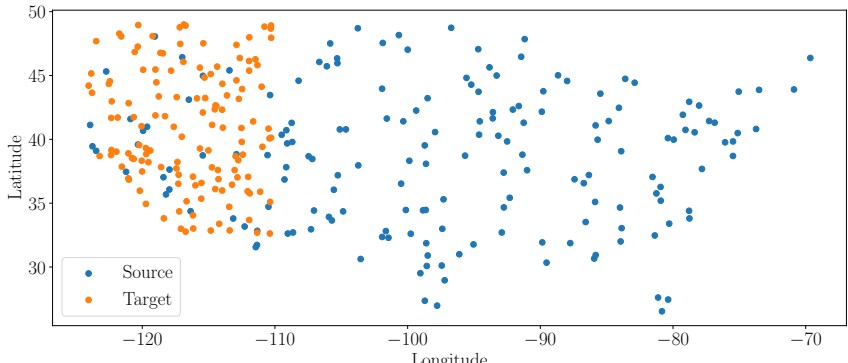

Figure 7: Split of the tree cover dataset in a target distribution in the West of the United States. Target locations are shown in orange, while source locations are shown in blue.

surface in kilometers, to compute distances in kilometers for the Wasserstein-1 cost and the nearest neighbor weighting procedure.

### E.3 Estimating the Ground Truth Parameters to Evaluate Coverage

In order to evaluate coverage, we repeat the data subsampling process described above 250 times. Ideally, we would estimate the coverage as the proportion of these seeds in which the estimand $\theta^\star_{\mathrm{OLS},p}$ falls inside the confidence interval we construct for each method. However, we cannot evaluate $\theta^\star_{\mathrm{OLS},p}$ directly, even though we have access to the target responses. To account for sampling variability, we use our method and each baseline to construct a confidence interval for the difference,

$$\hat{\theta}^\star_p - \hat{\theta}_p \tag{34}$$

where $\hat{\theta}^\star_p = e_p^{\mathrm{T}}(X^{\star\mathrm{T}}X^\star)^{-1}X^{\star\mathrm{T}}Y^\star$ is the estimated parameter using the target data (which our method and baselines don't have access to) and $\hat{\theta}_p$ is the estimated parameter we compute with each method using the source responses. If this confidence interval contains 0, we count the method as having covered the true parameter, while if it doesn't we count the method as not having covered the true parameter. We estimate the variance of $\hat{\theta}^\star_p$ using the model-trusting standard errors $\hat{\sigma}^2 = \frac{1}{N-P}\sum_{n=1}^{N} r_n^2$ where $r_n$ are the residuals of the model fit on the training data. We expect this to inflate our estimate of the standard variance of the target OLS estimate if the response surface is nonlinear, as the residuals will be larger due to bias. By possibly overestimating and incorporating this sampling variability into confidence intervals, we expect the calculated coverages to overestimate the true coverages. The resulting coverages are shown in the top row of Fig. 3.

In Fig. 8 we also report confidence intervals by calculating the proportion of times that $\hat{\theta}^\star_p$ is contained in each confidence interval. $\hat{\theta}_p$ is an unbiased estimate for $\theta^\star_{\mathrm{OLS},p}$ whether or not the model is well-specified. However, we might expect that the coverages reported with this approach underestimate the actual coverage of each method's confidence intervals due to not accounting for sampling variability in $\hat{\theta}^\star_p$.

### E.4 Experiment with Varying Lipschitz Constant

In this section, we provide an additional experiment with these real data where the focus is to assess how varying the underlying Lipschitz constant in Assumption 4 changes coverage and interval width. For this experiment we consider 9 different values for the Lipschitz constant, $L \in \{0.001, 0.005, 0.01, 0.02, 0.05, 0.1, 0.2, 0.5, 1\}$. This range of values is a reasonable range of values that a practitioner with domain knowledge in the tree cover field might be willing to specify. Indeed, the smallest Lipschitz constant considered, 0.001, corresponds to assuming that to have a 1% increase in tree cover we need to move $1/0.001 = 1000$ kilometers. This is quite an extreme value, but in some parts of arid regions such as New Mexico and Arizona it can be true. On the other hand,

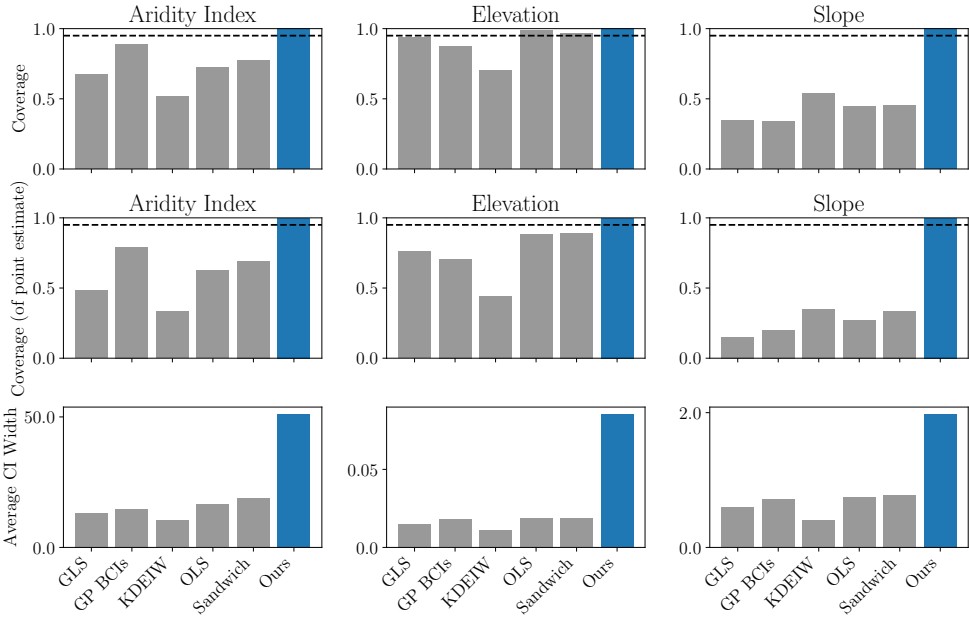

Figure 8: Coverages for the difference (upper), coverages for the point estimate (middle), and confidence interval widths (lower) for our method as well as 5 other methods for the West US data. Each column represents a parameter in the tree cover experiment. Only our method consistently achieves the nominal coverage.

the largest Lipschitz constant considered, 1, corresponds to assuming that to have $1\%$ increase in tree cover we need to move 1 kilometer. This is also extreme, but in regions in the US where elevation changes are very pronounced (e.g. in the Colorado Rockies), tree cover can vary sharply over very short distances.

In Fig. 9, we present the results of this experiment. We find that varying the Lipschitz constant within this range does not significantly impact coverage. Specifically, for *slope* and *elevation*, coverage remains consistent across all constants except $L = 0.001$. For *aridity index*, coverage is low for $L \leq 0.1$ but exceeds $95\%$ for $L \geq 0.2$. Meanwhile, confidence intervals become noticeably wider for $L > 0.5$ while remaining relatively stable for smaller values.

### E.5 Additional Experiment: Target South-East US

In this experiment, we define our target region in the Southeastern portion of CONUS at locations with latitude in the range (25, 38) and longitude in the range (-100, -75). Out of all spatial points in this region, $50\%$ — totaling 118 sites — are designated as target data. Next, we select the source data by taking a uniform random sample of $20\%$ of the remaining spatial locations, repeated over 250 random seeds to assess coverage performance. Each seed yields 173 source locations. Fig. 11 illustrates the spatial split between source and target data for a representative seed. As before, as a preprocessing step we convert the (latitude, longitude) coordinates of each data point into radians.

**Results.** We report the results for the confidence interval coverage and width in Fig. 12. As before, our method consistently achieves or exceeds $95\%$ nominal coverage for all parameters. All competing methods but KDEIW here achieves $95\%$ nominal coverage (or close to $95\%$ nominal coverage) for the aridity index and slope parameter when considering the coverage for the $\hat{\theta}_p^\star - \hat{\theta}_p$ (top row). All competing methods fall short of the nominal threshold for the elevation parameter. The wider

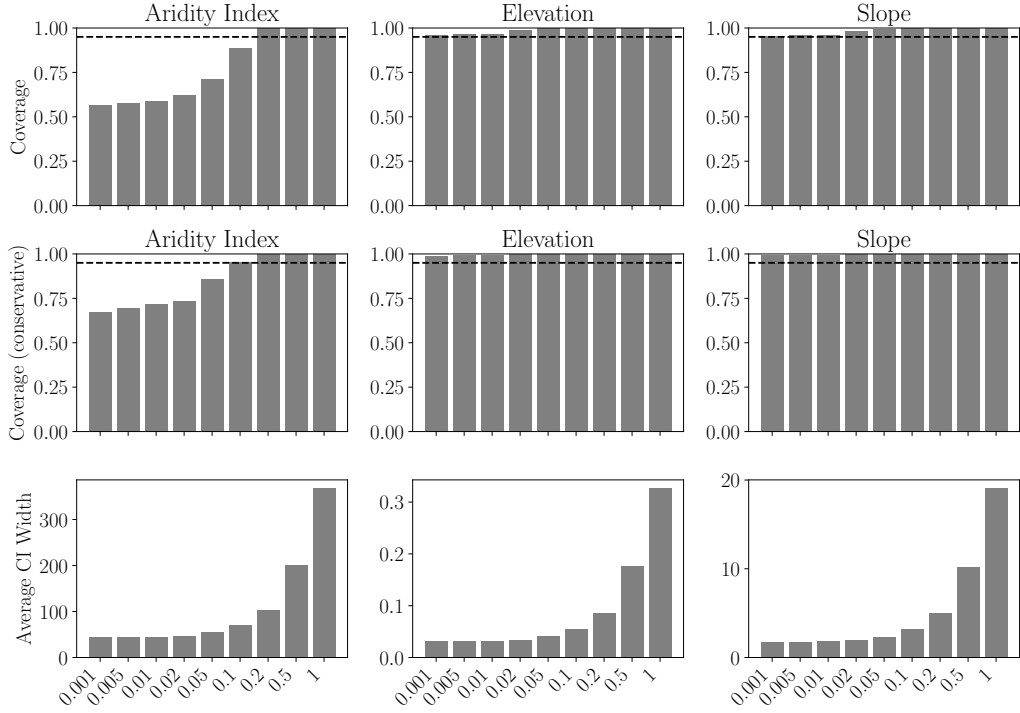

Figure 9: Coverage and average confidence interval widths over 250 seeds for 9 different values for the Lipschitz constant $L \in \{0.001, 0.005, 0.01, 0.02, 0.05, 0.1, 0.2, 0.5, 1\}$. The horizontal axis is shared across all plots.

intervals produced by our method (last row) reflect the trade-off between achieving reliable coverage and maintaining narrower intervals.

In the middle row we show the coverage for the point estimate $\hat{\theta}_p^\star$. Here we see how our method is the only one that achieves nominal coverage for all the parameters. In particular, all the other methods do not achieve nominal coverage for any of the parameters.

**Varying Lipschitz constant.** Finally, we report also for this experiment results when varying the assumed Lipschitz constant. As explained in Appendix E.4, we consider 9 different values for the Lipschitz constant, $L \in \{0.001, 0.005, 0.01, 0.02, 0.05, 0.1, 0.2, 0.5, 1\}$. We report the results in Fig. 14. As before, we find that varying the Lipschitz constant within this range does not significantly impact coverage. Specifically, here we see that for *aridity index* and *elevation*, coverage remains consistent across all constants above $L = 0.01$. For *slope*, 95% nominal coverage is achieved for $L \geq 0.05$. And — as before — confidence intervals become noticeably wider for $L > 0.5$ while remaining relatively stable for smaller values.

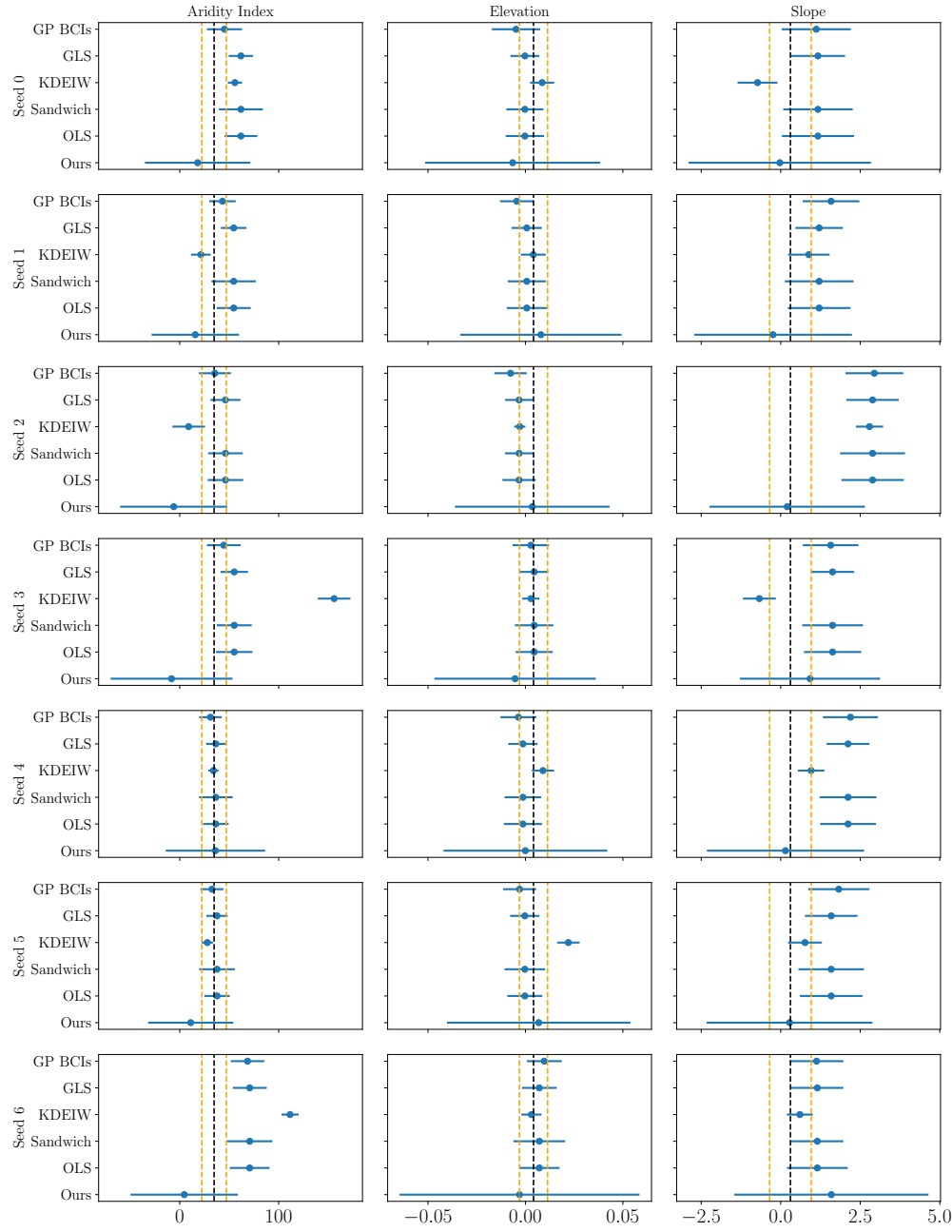

Figure 10: Confidence intervals for different seeds for the West US. Each row shows confidence intervals for the various methods over the three parameters for a given seed. The dashed vertical lines represent the true parameters (black is the point estimate, orange is a 95% confidence interval). The blue dots the point estimates for the different methods, and the blue lines are the confidence intervals.

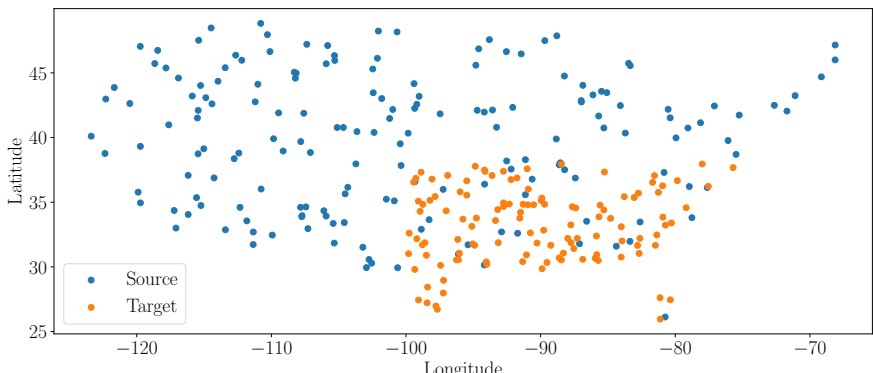

Figure 11: Spatial sites for the source (blue) and target (orange) data. The target data are chosen from the south-eastern part of the CONUS, whereas the source data cover the whole region.

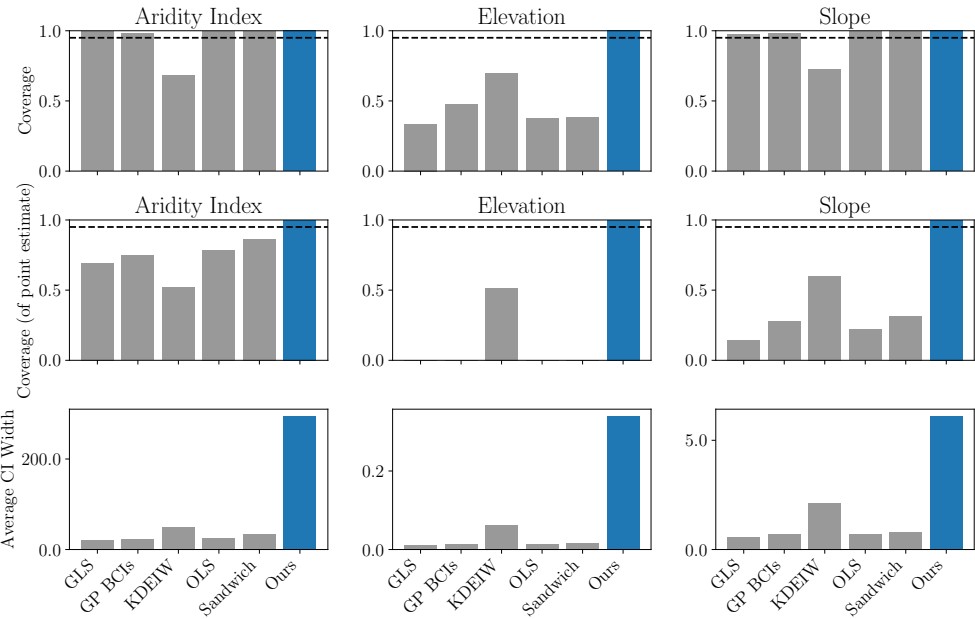

Figure 12: Coverages for the difference (upper), coverages for the point estimate (middle), and confidence interval widths (lower) for our method as well as 5 other methods for the Southeast US data. Each column represents a parameter in the tree cover experiment. Only our method consistently achieves the nominal coverage.

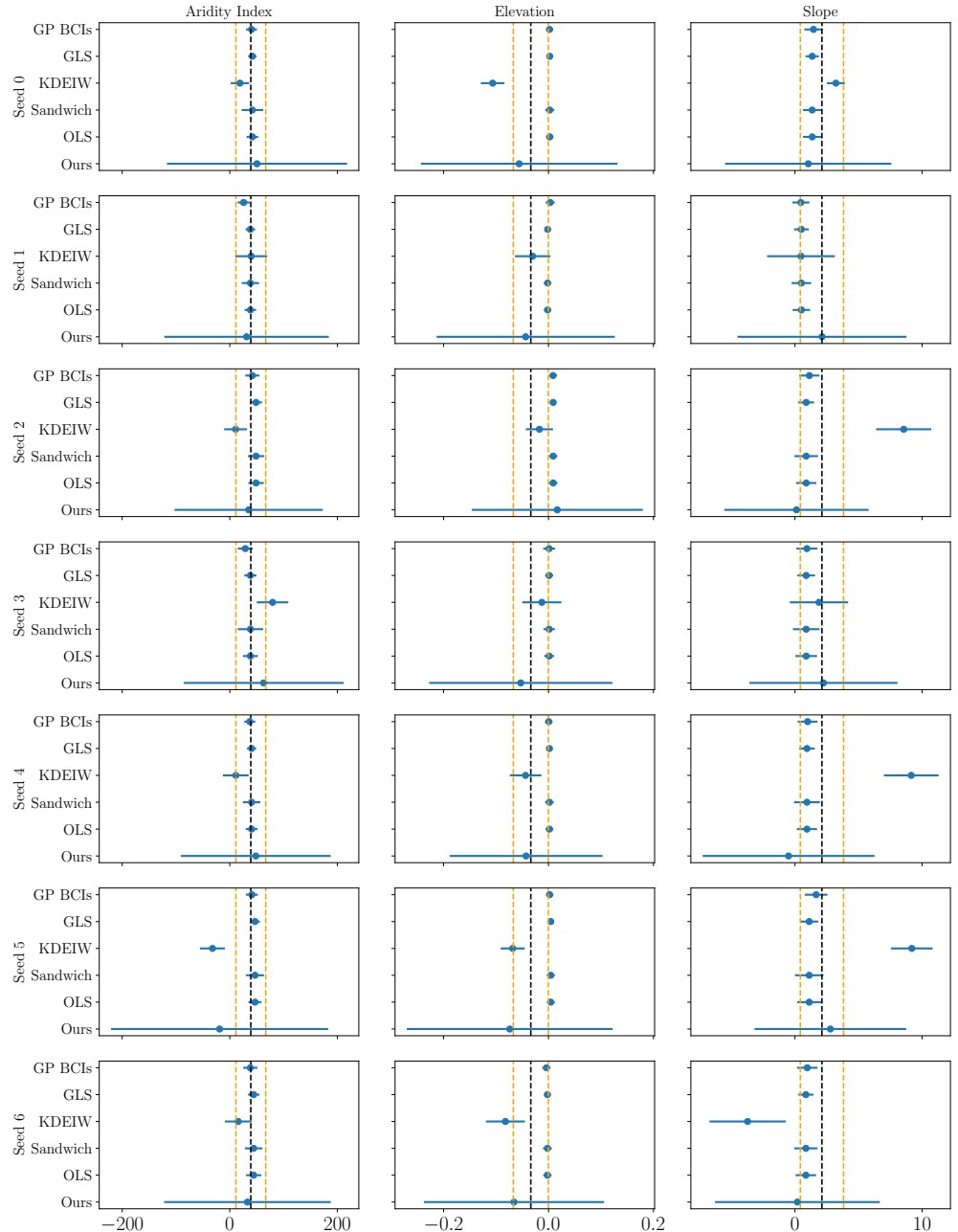

Figure 13: Confidence intervals for different seeds for the South East US. Each row shows confidence intervals for the various methods over the three parameters for a given seed. The dashed vertical lines represent the true parameters (black is the point estimate, orange is a 95% confidence interval). The blue dots the point estimates for the different methods, and the blue lines are the confidence intervals.

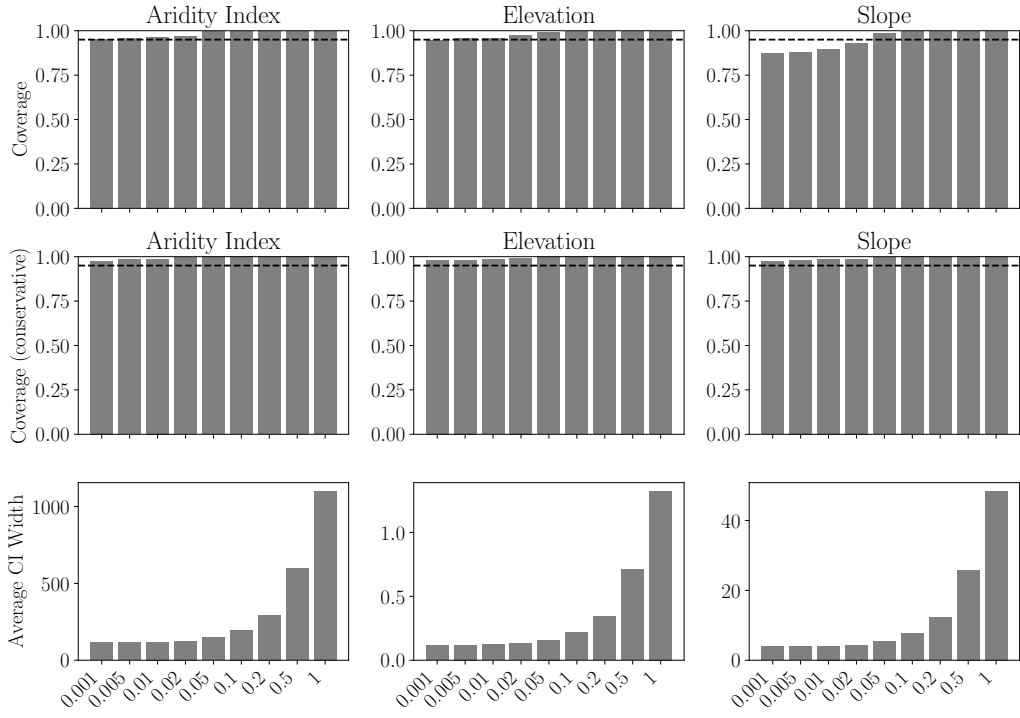

Figure 14: Coverage and average confidence interval widths over 250 seeds for 9 different values for the Lipschitz constant $L \in \{0.001, 0.005, 0.01, 0.02, 0.05, 0.1, 0.2, 0.5, 1\}$.

# F  Proofs

## F.1  Derivation of Target-Conditional Ordinary Least Squares

We derive the OLS (Ordinary Least Squares) estimand for the given optimization problem:

$$\theta_{\text{OLS}}^{\star} = \arg\min_{\theta \in \mathbb{R}^P} \mathbb{E}\left[\sum_{m=1}^{M}(Y_m^{\star} - \theta^{\top}X_m^{\star})^2 \Big| S_m^{\star}\right], \tag{35}$$

where $Y_m^{\star}$ is unobserved, and $X_m^{\star}$ is observed and a fixed function of $S_m^{\star}$.

First, expand the squared term inside the expectation:

$$(Y_m^{\star} - \theta^{\top}X_m^{\star})^2 = (Y_m^{\star})^2 - 2Y_m^{\star}\theta^{\top}X_m^{\star} + (\theta^{\top}X_m^{\star})^2. \tag{36}$$

Substitute this back into the expectation:

$$\mathbb{E}\left[\sum_{m=1}^{M}(Y_m^{\star} - \theta^{\top}X_m^{\star})^2\Big|S_m^{\star}\right] = \mathbb{E}\left[\sum_{m=1}^{M}(Y_m^{\star})^2 - 2Y_m^{\star}\theta^{\top}X_m^{\star} + (\theta^{\top}X_m^{\star})^2\Big|S_m^{\star}\right] \tag{37}$$

Since the expectation is linear, we can separate the terms:

$$\mathbb{E}\left[\sum_{m=1}^{M}(Y_m^{\star})^2\Big|S_m^{\star}\right] - 2\mathbb{E}\left[\sum_{m=1}^{M}Y_m^{\star}\theta^{\top}X_m^{\star}\Big|S_m^{\star}\right] + \mathbb{E}\left[\sum_{m=1}^{M}(\theta^{\top}X_m^{\star})^2\Big|S_m^{\star}\right] \tag{38}$$

Now we can simplify each of the terms as follows:

- The first term, $\sum_{m=1}^{M}\mathbb{E}\left[(Y_m^{\star})^2\Big|S_m^{\star}\right]$, does not depend on $\theta$, so it can be treated as a constant with respect to the optimization problem

- The second term, $-2\mathbb{E}\left[\sum_{m=1}^{M}Y_m^{\star}\theta^{\top}X_m^{\star}\Big|S_m^{\star}\right]$, can be rewritten using the linearity of expectation, the fact that $\theta$ is not random, and Assumption 1:

$$-2\theta^{\top}\sum_{m=1}^{M}X_m^{\star}\mathbb{E}\left[Y_m^{\star}\Big|S_m^{\star}\right] \tag{39}$$

- The third term, $\mathbb{E}\left[\sum_{m=1}^{M}(\theta^{\top}X_m^{\star})^2\Big|S_m^{\star}\right]$, is non-random by Assumption 1 and can be rewritten as:

$$\sum_{m=1}^{M}(\theta^{\top}X_m^{\star})^2 \tag{40}$$

The optimization problem then becomes

$$\theta_{\text{OLS}}^{\star} = \arg\min_{\theta \in \mathbb{R}^P}\left\{\text{constant} - 2\theta^{\top}\sum_{m=1}^{M}X_m^{\star}\mathbb{E}\left[Y_m^{\star}\Big|S_m^{\star}\right] + \sum_{m=1}^{M}(\theta^{\top}X_m^{\star})^2\right\}. \tag{41}$$

And since the constant term does not affect the optimization, we can drop it to get

$$\theta_{\text{OLS}}^{\star} = \arg\min_{\theta \in \mathbb{R}^P}\left\{-2\theta^{\top}\sum_{m=1}^{M}X_m^{\star}\mathbb{E}\left[Y_m^{\star}\Big|S_m^{\star}\right] + \sum_{m=1}^{M}(\theta^{\top}X_m^{\star})^2\right\} \tag{42}$$

To find the minimizer, we take the derivative of the objective function with respect to $\theta$ and set it to zero:

$$\frac{\partial}{\partial\theta}\left\{-2\theta^{\top}\sum_{m=1}^{M}X_m^{\star}\mathbb{E}\left[Y_m^{\star}\Big|S_m^{\star}\right] + \sum_{m=1}^{M}(\theta^{\top}X_m^{\star})^2\right\} = 0 \tag{43}$$

$$-2\sum_{m=1}^{M}X_m^{\star}\mathbb{E}\left[Y_m^{\star}\Big|S_m^{\star}\right] + 2\sum_{m=1}^{M}X_m^{\star}(X_m^{\star})^{\top}\theta = 0 \tag{44}$$

And by inverting this and using the fact that we assumed that $X^{\star\mathrm{T}}X^{\star}$ is invertible

$$\theta_{\mathrm{OLS}}^{\star} = \left(\sum_{m=1}^{M} X_m^{\star}(X_m^{\star})^{\top}\theta\right)^{-1}\sum_{m=1}^{M} X_m^{\star}\mathbb{E}\left[Y_m^{\star}\Big|S_m^{\star}\right] \tag{45}$$

which in matrix form can be written as

$$\theta_{\mathrm{OLS}}^{\star} = (X^{\star\mathrm{T}}X^{\star})^{-1}X^{\star\mathrm{T}}\mathbb{E}[Y^{\star}|S^{\star}]. \tag{46}$$

as in Eq. (4).

## F.2 Proof of Theorem 7

In the main text, we stated our confidence interval for the 1NN choice of $\Psi$. We now state version for general non-negative matrices with columns summing to 1.

**Theorem 11.** *Suppose $(S_m^{\star}, X_m^{\star}, Y_m^{\star})_{m=1}^{M}$ and $(S_n, X_n, Y_n)_{n=1}^{N}$ satisfy Assumptions 1 to 4. Define $w$ as in Algorithm 1. For any $\Psi \in \mathbb{R}^{M \times N}$, a matrix with non-negative entries with columns summing to 1, define $v^{\Psi} = w\Psi$. As in the main text, take $\hat{\theta}_p^{\Psi} = e_p^{\mathrm{T}}\left(X^{\star\mathrm{T}}X^{\star}\right)^{-1}X^{\star\mathrm{T}}\Psi Y$. Define $c = \sigma\|v^{\Psi}\|_2$, with $\sigma^2$ the variance of the additive noise from Assumption 2. Define the (random) interval $I^{\Psi}$ as in Algorithm 1 with known $\sigma^2$. Then with probability at least $1 - \alpha$, $\theta_{OLS,p}^{\star} \in I^{\Psi}$. That is, $I^{\Psi}$ has coverage (conditional on the test locations) at least $1 - \alpha$.*

*Proof.* We being as in the main text. We show the difference between our estimand and estimator is normally distributed. To that end, we decompose the difference between our estimand and estimator into a bias term and a mean-zero noise term.

$$\theta_{\mathrm{OLS},p}^{\star} - \hat{\theta}_p^{\Psi} = \sum_{m=1}^{M} w_m f(S_m^{\star}) - \sum_{n=1}^{N} v_n^{\Psi} Y_n, \tag{47}$$

for $w := e_p^{\mathrm{T}}\left(X^{\star\mathrm{T}}X^{\star}\right)^{-1}X^{\star\mathrm{T}} \in \mathbb{R}^M$ and $v^{\Psi} := w\Psi \in \mathbb{R}^N$. By Assumption 2, the righthand side of Eq. (47) can be written

$$\sum_{m=1}^{M} w_m f(S_m^{\star}) - \sum_{n=1}^{N} v_n^{\Psi} Y_n \underbrace{\sum_{m=1}^{M} w_m f(S_m^{\star}) - \sum_{n=1}^{N} v_n^{\Psi} f(S_n)}_{\text{bias}} - \underbrace{\sum_{n=1}^{N} v_n^{\Psi} \epsilon_n}_{\text{randomness}}. \tag{48}$$

Since the spatial locations are fixed, the bias term is not random and can be written as $b \in \mathbb{R}$. We can calculate the variance directly,

$$\mathbb{V}[\theta_{\mathrm{OLS},p}^{\star} - \hat{\theta}_p^{\Psi}] = \mathbb{V}[\sum_{n=1}^{N} v_n^{\Psi}\epsilon_n] = \sum_{n=1}^{N}(v_n^{\Psi})^2\mathbb{V}[\epsilon_n] = \sigma^2\|v^{\Psi}\|_2^2. \tag{49}$$

We used the $\epsilon_n$ are independent and identically distributed with variance $\sigma^2$ (Assumption 2).

It follows that

$$\theta_{\mathrm{OLS},p}^{\star} - \hat{\theta}_p^{\Psi} \sim \mathcal{N}(b, \sigma^2\|v^{\Psi}\|_2^2) \tag{50}$$

To bound the bias $b$, we use Assumption 4 to write

$$|b| \leq \sup_{g \in \mathcal{F}_L}\left|\sum_{m=1}^{M} w_m g(S_m^{\star}) - \sum_{n=1}^{N} v_n^{\Psi} g(S_n)\right|. \tag{51}$$

We can therefore apply Lemma 6 to complete the proof. $\qquad\square$

### F.3 Computing an Upper Bound on the Bias of Our Estimand with Wasserstein-1 Distance

First of all observe that if $\sum_{m=1}^{M} w_m = 1$, $\sum_{n=1}^{N} v_n^{\Psi} = 1$, $v_n^{\Psi} \geq 0$ for $1 \leq n \leq N$, and $w_m \geq 0$ for $1 \leq m \leq M$ then the supremum in Eq. (7) would be equal to a Wasserstein-1 distance by Kantorovich-Rubinstein duality [71, Theorem 5.10, Case 5.16].

Next, consider what happens if $\sum_{m=1}^{M} w_m - \sum_{n=1}^{N} v_n^{\Psi} \neq 0$. We show that the right-hand side of Eq. (7) is infinite. Assume by contradiction that there exists a $C > 0$ that upper bounds this supremum. Because $\sum_{m=1}^{M} w_m - \sum_{n=1}^{N} v_n^{\Psi} \neq 0$ and all constant functions are $L$-Lipschitz, for any $\gamma > 0$, taking, for all $S$, $g(S) = G = \frac{C+\gamma}{\sum_{m=1}^{M} w_m - \sum_{n=1}^{N} v_n^{\Psi}}$

$$\sup_{g \in \mathcal{F}_L} \left| \sum_{m=1}^{M} w_m g(S_m^{\star}) - \sum_{n=1}^{N} v_n^{\Psi} g(S_n) \right| \geq \left| \sum_{m=1}^{M} w_m G - \sum_{n=1}^{N} v_n^{\Psi} G \right|$$
$$= C + \gamma. \tag{52}$$

This contradicts the assumption that $C$ is an upper bound on the supremum. Because $C$ was arbitrary, the right hand side of Eq. (7) is infinite, as desired.

Our assumption that $\Psi$ is a non-negative matrix whose columns sum to 1 avoids this situation. We formalize our upper bound on the bias in the following proposition.

**Proposition 12.** *Suppose that* $\sum_{n=1}^{N} \Psi_{m,n} = 1$ *for all* $m$. *Let* $w \in \mathbb{R}^M$ *and* $v^{\Psi} = (w\Psi)^{\mathrm{T}} \in \mathbb{R}^N$.

$$\sup_{g \in \mathcal{F}_L} \left| \sum_{m=1}^{M} w_m g(S_m^{\star}) - \sum_{n=1}^{N} v_n^{\Psi} g(S_n) \right|$$
$$= ALW_1 \left( \sum_{m \in I} \frac{w_m}{A} \delta_{S_m^{\star}} + \sum_{n \in I'} \frac{-v_n^{\Psi}}{A} \delta_{S_n}, \sum_{m \in J} \frac{-w_m}{A} \delta_{S_m^{\star}} + \sum_{n \in J'} \frac{v_n^{\Psi}}{A} \delta_{S_n} \right), \tag{53}$$

*where* $I = \{1 \leq i \leq M : w_i \geq 0\}$, $I' = \{1 \leq i \leq N : v_i^{\Psi} < 0\}$, $J = \{1 \leq j \leq M : w_j < 0\}$ *and* $J' = \{1 \leq j \leq N : v_j^{\Psi} \geq 0\}$ *and* $A = \frac{1}{2} \left( \sum_{m=1}^{M} |w_m| + \sum_{n=1}^{N} |v_n^{\Psi}| \right)$.

*Proof.* First, observe that $\sum_{n=1}^{N} v_n^{\Psi} = \sum_{n=1}^{N} (w\Psi)_n = \sum_{n=1}^{N} \sum_{m=1}^{M} w_m \Psi_{m,n} = \sum_{m=1}^{M} w_m \sum_{n=1}^{N} \Psi_{m,n} = \sum_{m=1}^{M} w_m$.

Next, we normalize the weights to sum in absolute value to 2, and rescale the function class to consist of 1-Lipschitz function. Define $A = \frac{1}{2} \left( \sum_{m=1}^{M} |w_m| + \sum_{n=1}^{N} |v_n^{\Psi}| \right)$. Then

$$\sup_{g \in \mathcal{F}_L} \left| \sum_{m=1}^{M} w_m g(S_m^{\star}) - \sum_{n=1}^{N} v_n^{\Psi} g(S_n) \right| = \sup_{g \in \mathcal{F}_1} \left| \sum_{m=1}^{M} w_m L g(S_m^{\star}) - \sum_{n=1}^{N} v_n^{\Psi} L g(S_n) \right| \tag{54}$$

$$= AL \sup_{g \in \mathcal{F}_1} \left| \sum_{m=1}^{M} \frac{w_m}{A} g(S_m^{\star}) - \sum_{n=1}^{N} \frac{v_n^{\Psi}}{A} g(S_n) \right| \tag{55}$$

where we have used that a function is $L$-Lipschitz if and only if it can be written by scaling a 1-Lipschitz function by $L$.

Define $I = \{1 \leq i \leq M : w_i \geq 0\}$, $I' = \{1 \leq i \leq N : v_i^{\Psi} < 0\}$, $J = \{1 \leq j \leq M : w_j < 0\}$ and $J' = \{1 \leq j \leq N : v_j^{\Psi} \geq 0\}$. Then

$$AL \sup_{g \in \mathcal{F}_1} \left| \sum_{m=1}^{M} \frac{w_m}{A} g(S_m^{\star}) - \sum_{n=1}^{N} \frac{v_n^{\Psi}}{A} g(S_n) \right| \tag{56}$$

$$= AL \sup_{g \in \mathcal{F}_1} \left| \sum_{m \in I} \frac{w_m}{A} g(S_m^{\star}) + \sum_{n \in I'} \frac{-v_n^{\Psi}}{A} g(S_n) - \left( \sum_{m \in J} \frac{-w_m}{A} g(S_m^{\star}) + \sum_{n \in J'} \frac{v_n^{\Psi}}{A} g(S_n) \right) \right|$$
$$\tag{57}$$

Because $\sum_{n=1}^{N} v_n^{\Psi} = \sum_{m=1}^{M} w_m$ and $I$ and $J$ partition the index sets,

$$\sum_{m \in I} w_m + \sum_{n \in I'} -v_n^{\Psi} = \sum_{m \in J} -w_m + \sum_{n \in J'} v_n^{\Psi}. \tag{58}$$

And because the set $I, I', J, J'$ sort the indices into positive and negative parts

$$\sum_{m \in I} w_m + \sum_{m \in J} -w_m + \sum_{n \in I'} -v_n^{\Psi} + \sum_{n \in J'} v_n^{\Psi} = \sum_{m=1}^{M} |w_m| + \sum_{n=1}^{N} |v_n^{\Psi}| = 2A. \tag{59}$$

Therefore,

$$\sum_{m \in I} \frac{w_m}{A} \delta_{S_m^\star} + \sum_{n \in I'} \frac{-v_n^{\Psi}}{A} \delta_{S_n} \text{ and } \sum_{m \in J} \frac{-w_m}{A} \delta_{S_m^\star} + \sum_{n \in J'} \frac{v_n^{\Psi}}{A} \delta_{S_n} \tag{60}$$

are probability measures. We can apply Kantorovich-Rubinstein duality to write,

$$B \leq A L W_1 \left( \sum_{m \in I} \frac{w_m}{A} \delta_{S_m^\star} + \sum_{n \in I'} \frac{-v_n^{\Psi}}{A} \delta_{S_n}, \sum_{m \in J} \frac{-w_m}{A} \delta_{S_m^\star} + \sum_{n \in J'} \frac{v_n^{\Psi}}{A} \delta_{S_n} \right). \tag{61}$$

where $W_1$ denotes the 1-Wasserstein distance. $\qquad\square$

We compute the Wasserstein-1 distance by linear programming, see discussion in Appendix B.2. Scalable upper bounds could also be computed by exhibiting a coupling between the measures (for example by solving an entropy regularized optimal transport problem). See [49, Chapters 3 and 4] for details on computation of Wasserstein distances.

### F.4  Proof of Lemma 6

*Proof of Lemma 6.* We aim to prove that the interval $[-B - \Delta, B + \Delta]$ is the narrowest $1 - \alpha$ confidence interval that is valid for all $b \in [-B, B]$ where $\Delta$ is the solution of $\Phi(\Delta) - \Phi(-2B/\tilde{c} - \Delta) = 1 - \alpha$.

**Ensuring Coverage Probability.** Suppose that $X_b \sim N(b, \tilde{c}^2)$ for $b \in [-B - \Delta, B + \Delta]$. Then,

$$\Pr(X_b \in [-B - \Delta, B + \Delta]) = \Phi\left(\frac{B + \Delta - b}{\tilde{c}}\right) - \Phi\left(\frac{-B - \Delta - b}{\tilde{c}}\right),$$

To construct a valid confidence interval for any $b \in [-B, B]$, we require that

$$\Pr(X_b \in [-B - \Delta, B + \Delta]) \geq 1 - \alpha, \quad \forall b \in [-B, B].$$

This ensures $1 - \alpha$ coverage over all possible values of $b$ in $[-B, B]$.

**Reduce the problem to Worst-Case Coverage.** To find the narrowest interval, we identify the worst-case value of $b$ that minimizes the coverage probability. Let

$$C(b; \Delta) = \Phi\left(\frac{B + \Delta - b}{\tilde{c}}\right) - \Phi\left(\frac{-B - \Delta - b}{\tilde{c}}\right),$$

denote the coverage probability of the interval $[-B - \Delta, B + \Delta]$ for $X_b \sim \mathcal{N}(b, \tilde{c}^2)$. In order to ensure the interval is valid for all $b$ coverage, we want to bound below.

$$\inf_{b \in [-B, B]} C(b; \Delta)$$

The interval $[-B - \Delta, B + \Delta]$ is symmetric about 0, and the Probability Density Function for a Gaussian of mean $b$ is symmetric about $b$. Thus, the coverage probabilities at $b = -B$ and $b = B$ are equal. Consequently, it suffices to consider $b \in [0, B]$.

Moreover, observe that $C(b; \Delta)$ is a strictly decreasing function of $b$ on $[0, B]$ since (i) $\Phi\left(\frac{B + \Delta - b}{\tilde{c}}\right)$ decreases as $b$ increases (because $B + \Delta - b$ decreases and $\Phi(z)$ is monotonic) and (ii) $\Phi\left(\frac{-B - \Delta - b}{\tilde{c}}\right)$

also decreases as $b$ increases (because $-B - \Delta - b$ becomes more negative). Thus, $C(b; \Delta)$ is a strictly decreasing function of $b$ on $[0, B]$. The minimum value of $C(b; \Delta)$ occurs at $b = B$.

**Ensuring coverage in the worst case.** At the worst-case value $b = B$, the coverage probability is:

$$C(B; \Delta) = \Phi \left( \frac{\Delta}{\tilde{c}} \right) - \Phi \left( \frac{-2B - \Delta}{\tilde{c}} \right).$$

To ensure that the interval $[-B - \Delta, B + \Delta]$ achieves at least $1 - \alpha$ coverage for all $b \in [-B, B]$, we solve:

$$\Phi \left( \frac{\Delta}{\tilde{c}} \right) - \Phi \left( \frac{-2B - \Delta}{\tilde{c}} \right) = 1 - \alpha. \tag{62}$$

This guarantees that the interval is valid for all $b$ and achieves the desired coverage level.

**Narrowest interval.** The narrowest interval corresponds to the smallest $\Delta$ that satisfies the Eq. (62). By construction, any smaller $\Delta$ would fail to achieve the required coverage for $b = \pm B$, violating the validity condition. $\qquad \square$

### F.5   Proof of Proposition 8

In this section, we prove Proposition 8. For simplicity of exposition, we prove the result for $\mathcal{S} = [0, 1]^D$. The result generalizes to any spatial domain which is a compact metric space.

*Proof of Proposition 8.* Using Assumption 2 and expanding the quadratic form $(\epsilon_n + (f(S_n) - g(S_n)))$, we have

$$\hat{\sigma}_N^2 - \sigma^2 = Z_N + \zeta_N, \tag{63}$$

where $\zeta_N = \inf_{g \in \mathcal{F}_L} \frac{1}{N} \sum_{n=1}^{N} (f(S_n) - g(S_n))^2 + \frac{1}{N} \sum_{n=1}^{N} \epsilon_n (f(S_n) - g(S_n))$ and $Z_N = \frac{1}{N} \sum_{n=1}^{N} \epsilon_n^2 - \sigma^2$. Since $Z_N$ is an average of independent and identically distributed variable, and since $\mathbb{E}[Z_N] = 0$, the law of large numbers (LLN) implies $Z_N \to 0$ in probability. Because $Z_N \to 0$, if $\zeta_N \to 0$ in probability we can conclude by Slutsky's Lemma that $\hat{\sigma}_N^2 \to \sigma^2$ in probability. Therefore, the remainder of the proof involves showing $\zeta_N \to 0$ in probability.

Define $f_N$ to be the empirically centered version of $f$, that is $f_N = f - \frac{1}{N} \sum_{n=1}^{N} f(S_n)$. Then since the space of Lipschitz functions is invariant to shifts by constant functions

$$\zeta_N = \inf_{g \in \mathcal{F}_L} \frac{1}{N} \sum_{n=1}^{N} (f_N(S_n) - g(S_n))^2 + \frac{1}{N} \sum_{n=1}^{N} \epsilon_n (f_N(S_n) - g(S_n)). \tag{64}$$

Define the process,

$$\tau_N(g) = \frac{1}{N} \sum_{n=1}^{N} (f_N(S_n) - g(S_n))^2 + \frac{1}{N} \sum_{n=1}^{N} \epsilon_n (f_N(S_n) - g(S_n)), \tag{65}$$

so that $\zeta_N$ is the infimum of $\tau_N$. $\tau_N(f_N) = 0$. Therefore, for $\zeta_N \leq 0$ almost surely. It remains to show that for any $\delta < 0$, $\lim_{N \to \infty} \Pr(\zeta_N < \delta) \to 0$.

The essential challenge to showing that for any $\delta < 0$, $\lim_{N \to \infty} \Pr(\zeta_N < \delta) \to 0$ is the infimum over the space of Lipschitz functions. Our proof has three steps. First, we show that it suffices to consider a subset of the space of Lipschitz functions with bounded infinity norm. Second, we show that this space is compact as a subset of $L^\infty$. Third, we show that because the infimum is then over a compact set, it can be well-approximated by a minimum over a finite set of functions. And then a union bound suffices to prove the claim.

**Step 1: It's Enough to Consider a Bounded Subset of Lipschitz Functions.**

Because $f_N$ has empirical mean 0 and is continuous, by the intermediate value theorem it takes on the value 0 somewhere on $[0, 1]^D$. Because $f_N$ is $L$-Lipschitz and defined on a set of diameter $\sqrt{D}$, and 0 somewhere inside this set, $\|f_N\|_\infty \leq L\sqrt{D}$.

Define the set $\overline{\mathcal{F}_L} = \mathcal{F}_L \cap B_\infty(2L\sqrt{D} + 2\sigma^2)$, where $B_\infty(r)$ denotes the space of functions uniformly bounded by constant $r$ on $[0,1]^D$. By subadditivity of measure, for any $\delta < 0$

$$\Pr(\zeta_N < \delta) \leq \Pr\left(\inf_{g \in \overline{\mathcal{F}_L}} \tau_N(g) < \delta\right) + \Pr\left(\inf_{g \in \mathcal{F}_L \setminus \overline{\mathcal{F}_L}} \tau_N(g) < \delta\right) \tag{66}$$

$$\leq \Pr\left(\inf_{g \in \overline{\mathcal{F}_L}} \tau_N(g) < \delta\right) + \Pr\left(\inf_{g \in \mathcal{F}_L \setminus \overline{\mathcal{F}_L}} \tau_N(g) < 0\right). \tag{67}$$

We first consider the second term in this sum and show it tends to 0. We apply a crude Cauchy-Schwarz bound to the second term in Eq. (65) so that for any $g$,

$$\tau_N(g) \geq \sqrt{\frac{1}{N} \sum_{n=1}^{N} (f(S_n) - g(S_n))^2} \left(\sqrt{\frac{1}{N} \sum_{n=1}^{N} (f(S_n) - g(S_n))^2} - \sqrt{\frac{1}{N} \sum_{n=1}^{N} \epsilon_n^2}\right). \tag{68}$$

Therefore, $\tau(g) < 0$ implies

$$\frac{1}{N} \sum_{n=1}^{N} \epsilon_n^2 \leq \frac{1}{N} \sum_{n=1}^{N} (f_N(S_n) - g(S_n))^2. \tag{69}$$

For any $g \in \mathcal{F}_L \setminus \overline{\mathcal{F}_L}$ because $g$ takes on a value at least $2L\sqrt{D} + 2\sigma^2$ and is $L$-Lipschitz, $g$ is larger than $L\sqrt{D} + 2\sigma^2$ over the entire unit cube. And because $\|f_N\| \leq L\sqrt{D}$, $f_N(S_n) - g(S_n) \geq 2\sigma^2$ for all $n$. Therefore for all $g \in \mathcal{F}_L \setminus \overline{\mathcal{F}_L}$

$$\frac{1}{N} \sum_{n=1}^{N} (f_N(S_n) - g(S_n))^2 \geq 2\sigma^2. \tag{70}$$

We conclude that

$$\lim_{N \to \infty} \Pr\left(\inf_{g \in \mathcal{F}_L \setminus \overline{\mathcal{F}_L}} \tau_N(g) < 0\right) \leq \lim_{N \to \infty} \Pr\left(\frac{1}{N} \sum_{n=1}^{N} \epsilon_n^2 \geq 2\sigma^2\right) = 0. \tag{71}$$

where the final inequality is the law of large numbers.

**$\overline{\mathcal{F}_L}$ is a Compact Subset of the Space of Bounded Functions with Sup Norm.** All that remains to show is that for any $\delta < 0$,

$$\Pr\left(\inf_{g \in \overline{\mathcal{F}_L}} \tau_N(g) < \delta\right) \to 0. \tag{72}$$

The idea (following standard arguments made with empirical processes) is that we can take a cover of $\overline{\mathcal{F}_L}$ of finite size, such that each element of $\tau_N(g)$ is almost constant over elements of this cover. This essentially lets us approximate the infimum with a minimum over a finite set, up to small error. And once the problem is reduced to a minimum we can apply a union bound and the law of large number. We will formalize this in the next paragraph, but we first show that $\overline{\mathcal{F}_L}$ is compact.

Because every Lipschitz continuous function is equicontinuous, functions in $\overline{\mathcal{F}_L}$ are pointwise bounded by $2L\sqrt{D} + 2\sigma^2$, and $[0,1]^D$ is compact, we may apply Arzela-Ascoli [58, Theorem 7.25] to conclude that $\overline{\mathcal{F}_L}$ with the sup norm is sequentially compact. It is therefore compact, as a sequentially compact metric space is compact.

**Step 3: Reduction to a Minimum over a Finite Set and a Union Bound.**

For any $\delta' < 0$ and any $g \in \overline{\mathcal{F}_L}$,

$$\lim_{N \to \infty} \Pr(\tau_N(g) < \delta') \leq \lim_{N \to \infty} \Pr\left(\frac{1}{N} \sum_{n=1}^{N} \epsilon_n g(S_n) < \delta'\right) = 0 \tag{73}$$

where the final equality is the law of large numbers.

Therefore, for any finite set $C \subset \overline{\mathcal{F}_L}$,

$$\lim_{N \to \infty} \Pr(\min_{g \in C} \tau_N(g) < \delta') \leq \lim_{N \to \infty} \sum_{g \in C} \Pr\left(\frac{1}{N} \sum_{n=1}^{N} \epsilon_n g(S_n) < \delta'\right) \tag{74}$$

$$= \sum_{g \in C} \lim_{N \to \infty} \Pr\left(\frac{1}{N} \sum_{n=1}^{N} \epsilon_n g(S_n) < \delta'\right) = 0. \tag{75}$$

We used countable subadditivity in the inequality.

Now for any $\gamma > 0$, there exists a finite set of functions $C_\gamma \subset \overline{\mathcal{F}_L}$ such that for any $g \in \overline{\mathcal{F}_L}$, there exists a $g' \in C_\gamma$ with $\|g - g'\|_\infty \leq \gamma$. And since $\tau_N(g)$ is pathwise uniformly (in $N$) continuous on $\overline{\mathcal{F}_L}$ equipped with sup norm,

$$\inf_{g \in \overline{\mathcal{F}_L}} \tau_N(g) \geq \min_{g \in C_\gamma} \tau_N(g) - \rho(\gamma) \tag{76}$$

where is a nonnegative function such that $\lim_{\gamma \to 0} \rho(\gamma) = 0$. Therefore, for any $\delta < 0$, we can find a $\gamma$ such that $\rho(\gamma) \leq -\frac{\delta}{2}$. For this $\gamma$, applying Eq. (74) with $\delta' = \frac{\delta}{2}$ allows us to conclude that

$$\lim_{N \to \infty} \Pr(\inf_{g \in \overline{\mathcal{F}_L}} \tau_N(g) \leq \delta) = 0. \tag{77}$$

This is a uniform law of large number for the class of Lipschitz and bounded functions. More quantitative results are likely possible using empirical process theory, see Wainright [73, Chapter 5]. □

**Proof of Asymptotic Coverage (Corollary 9).**  We now prove Corollary 9. We begin by recalling the definition of an asymptotically valid confidence interval.

**Definition 13.** *We say a sequence of (random) intervals $(I_N)_{N=1}^\infty$ has asymptotically valid coverage of $\theta$ at level $(1 - \alpha)$ if*

$$\lim_{N \to \infty} \Pr(\theta \in I_N) = 1 - \alpha \tag{78}$$

Corollary 9 follows from Theorem 7 and Proposition 8 by the following lemma, which is a special case of Slutsky's lemma.

**Lemma 14.** *Let $\sigma^2 > 0$. Suppose that $\theta^\star_{OLS,p} - \hat{\theta}_p^\Psi - b_N \sim \mathcal{N}(0, \sigma^2)$ where $(b_N)_{N=1}^N$ is a fixed sequence. Suppose $\hat{\sigma}_N^2$ converges in probability to $\sigma^2$. Then,*

$$\frac{1}{\hat{\sigma}_N^2}\left(\theta^\star_{OLS,p} - \hat{\theta}_p^\Psi - b_N\right) \to \mathcal{N}(0, 1) \tag{79}$$

*where convergence is in probability.*

*Proof.* The result is a special case of Slutsky's lemma, using $\hat{\sigma}_N^2 \to \sigma^2 > 0$. □

*Proof of Corollary 9.* Define $b_N = \sum_{m=1}^{M} w_m f(S_m^\star) - \sum_{n=1}^{N} v_n^\Psi f(S_n^\star)$. Because $\theta^\star_{OLS,p} - \hat{\theta}_p^\Psi \sim \mathcal{N}(b, \sigma^2\|v^\Psi\|_2^2)$,

$$(\theta^\star_{OLS} - \hat{\theta}_p^\Psi - b_N) \sim \mathcal{N}(0, \sigma^2). \tag{80}$$

If $\sigma^2 = 0$, then $\hat{\sigma}_N^2 = 0$ for all $N$, because the conditional expectation is an $L$-Lipschitz function leading to 0 squared error in the minimization algorithm used to calculate $\hat{\sigma}_N^2$. Therefore, the resulting confidence interval has coverage $(1 - \alpha)$ for all $N$ by Theorem 7.

For $\sigma^2 > 0$ we apply Lemma 14 to conclude that

$$\frac{1}{\hat{\sigma}_N^2}\left(\theta^\star_{OLS,p} - \hat{\theta}_p^\Psi - b_N\right) \to \mathcal{N}(0, 1) \tag{81}$$

where convergence is in probability.

Convergence in probability implies convergence in distribution, and therefore convergence of quantiles at continuity points. The Gaussian CDF is continuous. Therefore, the quantile computation in Lemma 6 using $\hat{\sigma}_N^2$ in place of $\sigma^2$ produces an asymptotically valid confidence interval in the limit as $N \to \infty$.

□

