# OpenReview forum: "Smooth Sailing: Lipschitz-Driven Uncertainty Quantification for Spatial Associations"
_NeurIPS.cc/2025/Conference — NeurIPS 2025 poster_

### Official Review · Reviewer_NCSm · 2025-06-27

**Clarity:** 4
**Significance:** 3
**Originality:** 3
**Rating:** 4
**Confidence:** 3

**Summary:**

This paper proposes a new method for constructing valid confidence intervals for spatial associations under model misspecification and distribution shift. By assuming only that the response is Lipschitz-continuous in space, the method avoids strong modeling assumptions and achieves nominal coverage where existing methods fail. Experiments on simulated and real data show superior coverage and efficiency compared to standard approaches.

**Questions:**

1. The paper considers the setting where $Y$ is an $N$-long column vector. How would the proposed method extend to cases where $Y$ is high-dimensional (i.e., multivariate responses)?

2. From a mathematical perspective, what are the fundamental reasons that make the proposed method better than Gaussian Processes (GP BCI), particularly under model misspecification or distribution shift?

**Ethical Concerns:**

["NO or VERY MINOR ethics concerns only"]

**Final Justification:**

I would like to give score 4 for this paper.

**Limitations:**

yes

**Quality:**

4

**Strengths And Weaknesses:**

**Strengths**:
- **Clarity and Accessibility**: The paper is very clearly written and easy to follow. Even for readers outside the immediate subfield, the key ideas and contributions are accessible.
- **Strong Motivation**: The motivation is compelling and well-grounded in real-world challenges of spatial inference under model misspecification and distribution shift.
- **Theoretical Rigor**: The analysis is mathematically solid, with well-justified assumptions and detailed, carefully presented proofs.

**Weakness**:
- **Title Meaning**: Why is the paper titled *"Smooth Sailing"*? How does it relate to the method and its content?

---

> ### Author Rebuttal · Authors · 2025-07-31
>
> We thank the reviewer for their insightful feedback and thoughtful questions. We are excited to hear that the reviewer found our paper clearly written, strongly motivated, and mathematically rigorous. Below we address each question in turn.
>
> **Why “Smooth Sailing”?** The “Smooth Sailing” part of our title is meant to be a pun, playing on two meanings of “smoothness.” (1) “Smooth sailing” is an English idiom that denotes “easy progress without impediment or difficulty” (Merriam-Webster Dictionary). (2) Under only a Lipschitz-continuity assumption (a “smoothness” assumption), our procedure guarantees valid confidence intervals — even in the presence of model misspecification and covariate shift. So our goal was to convey that, under a smoothness (Lipschitz) assumption, inference using our procedure proceeds smoothly (easily).
>
> **Extension to multivariate responses.** While our paper focuses on a scalar response $Y(s)\in \mathrm{R}$, very similar machinery can be adapted to confidence intervals for each coordinate of $Y(s)\in \mathrm{R}^D$. Namely, we can treat each component $Y^{(d)}(s)$ as a separate univariate problem, and apply our Lipschitz‐based bias bound and variance calibration to each coordinate. If one desires simultaneous coverage over all $D$ outputs, a straightforward Bonferroni correction or any family‐wise error control can be used. We haven’t tested this proposal in practice, as our focus was on scalar responses, but we think it would be an interesting future task. We will add a short paragraph in Section 3 outlining this extension.
>
> **Why not Gaussian process credible intervals?** Gaussian process–based Bayesian credible intervals are popular in spatial statistics. As is always the case with Bayesian approaches, the credible intervals represent subjective beliefs. In the presence of misspecification (which we focus on in this work) posterior credible intervals can under or overestimate uncertainties about the parameter (Müller, 2013; Walker, 2013). This is even more the case when there is also a distribution shift, as the prior is likely to have a large impact on the posterior when we are extrapolating. In contrast, our method (i) explicitly bounds worst‐case bias via a 1-Wasserstein distance and (ii) adds this bias bound to the confidence interval, yielding guaranteed frequentist coverage under arbitrary covariate shift. In contrast, our method (i) explicitly bounds worst‐case bias via a 1-Wasserstein distance and (ii) adds this bias bound to the confidence interval, yielding guaranteed frequentist coverage under arbitrary covariate shift.
>
> We hope these clarifications address your questions.  Thank you again for your constructive feedback!
>
> **References:**
>
> Müller. “Risk of Bayesian inference in misspecified models, and the sandwich covariance matrix.” Econometrica.  2013.
>
> Walker. “Bayesian inference with misspecified models.” Journal of Statistical Planning & Inference. 2013.

---

> ### Comment · Reviewer_NCSm · 2025-08-04
>
> Thanks for your detailed response and I will maintain my current score.

---

### Official Review · Reviewer_jtMd · 2025-07-01

**Clarity:** 3
**Significance:** 2
**Originality:** 3
**Rating:** 4
**Confidence:** 1

**Summary:**

The paper concerns estimating associations between covariates and responses with the correct frequentist coverage of the confidence interval. With this aim, the authors proposed a new method for associations in spatial settings.

**Questions:**

please see the strength and weakness section.

**Ethical Concerns:**

["NO or VERY MINOR ethics concerns only"]

**Final Justification:**

Thanks to the authors for clarifying the aims of the paper and responding to my questions.
Now I have a better understanding of the paper, and I've changed my score.

**Limitations:**

yes

**Paper Formatting Concerns:**

NA.

**Quality:**

2

**Strengths And Weaknesses:**

The authors described the ideas and demonstrated them using simulation studies. I have some concerns about the novelty, although I might overlook it.
I struggled to understand the logic behind Assumptions 1 and 2, the existence of independent maps from spatial locations to the response and covariates, respectively. i.e., Two data-generating processes. It would be more natural to consider a parametric model with covariates which are mapped from locations to account for the association between response and spatial location. In the simulation studies, the authors generated data in this way. I think it is good to clarify this point.

Good estimation of a map between location and covariate influences the parameter estimation; the authors assume that they are smooth, indicated by the Lipschitz constant. To me, the smoothness assumption of this map sounds overly simplified, and there is a bias-variance trade-off and more thorough discussion and studies about its impact on parameter estimation.

Although the authors discussed variance parameter estimation in the paper, a confidence interval with the nominal coverage challenges, as the authors noted in the simulation study. Simple linear models are considered in the paper. Although the authors plan to extend this to GLM, the current version seems to present limited studies.

---

> ### Author Rebuttal · Authors · 2025-07-31
>
> We thank the reviewer for the feedback and comments. Before responding to individual points, we recall that our goal in the present paper is to provide valid confidence intervals for the signed strength of the association between covariates and a response in regression.
>
> **Lack of novelty and focus on linear regression.** Given later comments in the review, we interpret the reviewer’s concern about lack of novelty to stem from our focus on the analysis of linear regression. If the reviewer has other novelty concerns, we ask them to please clarify those.
>
> The novelty of our paper comes from recognizing and addressing biases introduced into estimation of associations in spatial settings; in particular, the biases due to model misspecification and distribution shift are ubiquitous in real-life problems. We provide the first framework for correcting these biases when reporting associations between covariates and a response.
>
> We would like to push back on the idea that a paper focused on linear regression cannot be novel or relevant to the ML community precisely because it is on the topic of linear regression. We first note that, as we argue in our introduction, linear regression just happens to be the current best tool (even considering more complex, recently-developed methods) for the particular data analysis task of quantifying associations (a task that is extremely widespread across the sciences and social science). So our focus is not on linear regression for its own sake but instead because of its appropriateness for the task. Indeed, there remain many tasks of interest to ML for which linear regression is an appropriate tool. For this reason, numerous papers were published even just at last year’s NeurIPS on topics related to linear regression (e.g. Xie and Huo 2024, Lin et al, 2024, Jain et al. 2024, Zhu et al. 2024, Liu and Novikov 2024).
>
> That all being said, we agree with the reviewer that extensions of our approach beyond linear regression are interesting. We hope that our present work will lay the groundwork for that future work, which we expect to require substantial additional effort.
>
> **Motivation for Assumptions 1 and 2.** (*“I struggled to understand the logic behind Assumptions 1 and 2, the existence of independent maps from spatial locations to the response and covariates, respectively. i.e., Two data-generating processes. It would be more natural to consider a parametric model with covariates which are mapped from locations to account for the association between response and spatial location. In the simulation studies, the authors generated data in this way. I think it is good to clarify this point.”*)
>
> We understand the reviewer to be suggesting that our assumptions (particularly Assumption 2) exclude the possibility that the response is a function of the covariate. We emphasize that, while in Assumption 2 we formally state that the response is a function of the spatial location, the response can also be a function of the covariates. We do not state this potential dependence explicitly in Assumption 2 because of Assumption 1. Formally, suppose that $E[Y|X, S] = g(X, S)$. By Assumption 1, $X = \chi(S)$, and so $E[Y|X,S] = g(\chi(S), S)$. Define $f: \mathcal{S} \to \mathbb{R}$ by $f(s) =  g(\chi(S), S)$. Then $E[Y|X,S] = E[Y|X] = g(\chi(S), S)$. In particular, the response can be a function of the covariates, but because the covariates are a fixed function of space, we can also write the response as a function of space alone. We think it is often easier to reason about smoothness in physical space, than in a (potentially high-dimensional) space of covariates. And so we impose our smoothness restrictions on the average response as a function of space alone, but this formulation does not mean the response cannot depend on the covariates as outlined above. We will clarify this point in our revision.
>
> We believe the reviewer is suggesting to instead assume the data arises according to $Y = f(X) +$ noise, with a parametric choice of f. Choosing a parametric model for f would require agreeing on an appropriate model. The linear model is a natural choice but recovers the well-specified case, which is both well-studied and unrealistic. Other parametric choices might be considered arbitrary and limiting. As we describe in the paragraph above, we are essentially already making a similar assumption, but with a nonparametric choice of f, and we allow the latent function to depend on the spatial location in additional ways beyond just via the covariates.
>
> **Smoothness of maps (functions) 1.** (*“Good estimation of a map between location and covariate influences the parameter estimation; the authors assume that they are smooth, indicated by the Lipschitz constant.”*)
>
> We understand the reviewer to be concerned with the problem of estimating a mapping between spatial locations and covariates. We would like to emphasize that, while we assume a map between spatial locations and covariates exists, we do not need to estimate it. Nor do we make any smoothness assumptions about it. We **do** make a smoothness assumption on the expected response.
>
> **Smoothness of maps (functions) 2.** (*“To me, the smoothness assumption of this map sounds overly simplified”*)
> While we reiterate that we do **not** require a Lipschitz assumption on the covariates as a function of the spatial locations, the reviewer might be concerned about the smoothness assumption we do make on the expected response. We chose to make this smoothness assumption because (1) it was simple enough to be physically interpretable and (2) there are real-data analyses where it is a reasonable assumption. For a concrete example of the second point, see our response to reviewer W9eu; there we reference a study that suggests concrete scales over which the annual average PM2.5 in California varies, and these could be used to select a Lipschitz constant of, for example 1.2 $\mu g/m^3$. We emphasize that any machine learning method must require some assumption about how much a response is allowed to vary as you move in space — so that the method can perform inference at locations where we haven’t directly observed data; if we allow arbitrarily large changes in the response within an arbitrarily small spatial distance, even seemingly simple interpolations become impossible. While one could choose a more complicated assumption, we worry that alternative assumptions could lose physical interpretability, making them difficult to use in practice.
>
> **Bias-variance trade-off.** (*“there is a bias-variance trade-off and more thorough discussion and studies about its impact on parameter estimation.”*)
> We assume the reviewer is concerned with the bias-variance tradeoff that arises from specifying a Lipschitz constant — i.e. that with our method for estimating the Lipschitz constant, a higher variance is estimated when a small Lipschitz constant is specified, and a smaller noise variance is estimated when a larger Lipschitz constant is specified. We illustrated the impact this trade-off has on parameter estimation in Figure 2 on simulated data, as well as on real-data in Figure 9 in the appendix. On the simulated data, we saw that confidence interval width is not always monotonically increasing with the Lipschitz constant, because of smaller estimates of the noise variance (Figure 2, middle and right). But when more extrapolation is required, larger Lipschitz constants lead to wider confidence intervals (Figure 2, left). In the real-data experiment, we also saw that for the dataset considered, larger Lipschitz constants led to wider confidence intervals. We further found that our method maintained nominal coverage across a wide-range of Lipschitz constants, suggesting that choosing a small Lipschitz constant in many cases still can correct bias. These observations support that attempting to account for the bias introduced by misspecification and distribution shift is preferable to ignoring it all together.
>
> **Extensions beyond linear models and limited studies.** (*“Although the authors plan to extend this to GLM, the current version seems to present limited studies.”*)
> While we agree the extension of our method to GLMs is of interest (indeed we are working on this direction), it requires non-trivial work beyond what is in this paper to ensure the confidence intervals are valid. In this paper, we developed a framework for addressing bias due to distribution shift and misspecification when estimating spatial associations using linear regression; our present work already required novel assumptions and technical arguments. To address the stated concern about limited studies, we note that we included several (3) simulation studies as well as two analyses on a real data set in the current paper. We feel that these illustrate the key benefits, considerations, and limitations of our method. If the reviewer has specific other studies in mind (or substantive points that would require additional studies to address), we would be grateful for suggestions.
>
> **References:**
>
> Chow, Chen, Watson, Lowenthal, Magliano, Turkiewicz, Lehrman. PM2.5 Chemical Composition and Spatiotemporal Variation During the California Regional PM10/PM2.5 Air Quality Study. Journal of Geophysical Research. 2006.
>
> Xie, Huo. High-dimensional (Group) Adversarial Training in Linear Regression. NeurIPS 2024.
>
> Lin, Wu, Kakade, Bartlett, Lee. Scaling Laws in Linear Regression: Compute, Parameters, and Data. NeurIPS 2024.
>
> Jain, Sen, Kong, Das, Orlitsky. Linear Regression using Heterogeneous Data Batches. NeurIPS 2024.
>
> Zhu, Manseur, Ding, Liu, Xu, Wang. Truthful High Dimensional Sparse Linear Regression. NeurIPS 2024.
>
> Chih-Hung Liu, Gleb Novikov. Robust Sparse Regression with Non-Isotropic Designs. NeurIPS 2024

---

> > ### Comment · Reviewer_jtMd · 2025-08-05
> >
> > Thanks to the authors for clarifying the aims of the paper and responding to my questions. Now I have a better understanding of the paper, and I've changed my score.

---

### Official Review · Reviewer_pd1Y · 2025-07-03

**Clarity:** 3
**Significance:** 3
**Originality:** 3
**Rating:** 4
**Confidence:** 2

**Summary:**

This paper proposes confidence intervals for association in spatial setting where the linear model is misspecified. Additive Gaussian noise assumption and covariates are fixed function of spatial location assumption are imposed on the data-generating process. Further, under the assumption that the response is a Lipschitz function of the spatial location is assumed, and the authors provide confidence intervals with valid coverage based on these conditions.

**Questions:**

In addition to the questions raised in above, I found the Assumption 1 very restrictive. It effectively washes away the need for covariates in this setting. Assumption 1 states that for fixed spatial location parameter in the model, the corresponding covariate value is always the same. I found this assumption unreasonable across many applications that the authors mentioned (e.g., air pollution). The explanation that the authors provided in the paper does not directly justify why this assumption holds in applications.

**Ethical Concerns:**

["NO or VERY MINOR ethics concerns only"]

**Final Justification:**

The provided justification by the authors further clarified the contributions of the paper. I was not able to check the mathematical details of the paper, so i will keep my score at 4.

**Limitations:**

Yes

**Quality:**

3

**Strengths And Weaknesses:**

The paper is well-organized, with clear logic between different sections. I didn’t check the mathematical details of the paper, but it seems solid at first glance. I do not work in the spatial association area, so it is hard for me to judge the contribution of the paper given the assumptions made in the paper.

I found the introduction poorly written. The second to the fourth paragraph talks about possible relationships between source and spatial location, and lacks focus. It is unclear what the authors are trying to convey. I would recommend the authors to focus directly on the linear model and discuss other possible models either in related work or discussion section.

I found the claim on the contribution of the paper in accurate: the authors claim that their result hold under non parametric assumptions on the data-generating distributions. However, as stated in their assumptions 1 and 2, additive Gaussian noise assumption is clearly imposed on their data-generating process. Additionally, they require smoothness assumptions on the DGP. The statement is very misleading.

---

> ### Author Rebuttal · Authors · 2025-07-31
>
> We thank the reviewer for their thoughtful feedback. We are glad to hear they found our paper well-organized and mathematically rigorous. Before responding to individual points, we recall that our goal in the present paper is to provide valid confidence intervals for the signed strength of the association between covariates and a response in regression.
>
> **Purpose of paragraphs 2–4 in the introduction.** The reviewer felt that paragraphs 2–4 of the introduction lacked focus. An important premise for our work is that linear regression is still a useful tool, even when the linear model is misspecified (which is always the case). We think making an explicit argument for the relevance of linear regression is particularly important given that the NeurIPS community often emphasizes algorithmic and computational approaches to prediction problems, over model-based approaches. The goal of these paragraphs, and the introduction, is to describe why the reader should care about the problem we study, before getting into the assumptions we make and our proposed solution.
>
> **Use of the term nonparametric.** The reviewer suggests that, because we assume additive Gaussian noise and make a smoothness assumption on the latent function, we should not call our method nonparametric. We believe our use of the term *nonparametric* in the present work follows standard usage in the machine learning literature: namely, we use it to describe a model that is not completely specified in terms of a finite-dimensional parameter vector. For example, Gaussian process regression is generally referred to as nonparametric, even when the latent function is smooth and the observation noise is i.i.d. Gaussian (Giordano et al, 2022, Zhang et al, 2022; El Harzli et al 2024). And papers analyzing kernel ridge regression refer to the method as nonparametric, even if the analysis is done assuming i.i.d. Gaussian noise and a smooth latent function (Cui et al, 2021). As Michael Jordan (2010) wrote, “The word ‘nonparametrics’ needs a bit of explanation. The word does not mean ‘no parameters’; indeed, many stochastic processes can be usefully viewed in terms of parameters (often, infinite collections of parameters). Rather, it means ‘not parametric,’ in the sense that [...] nonparametric inference is not restricted to objects whose dimensionality stays fixed as more data is observed.” We contrast this usage with the term *distribution-free inference* (Cherrian, 2024; Correia 2024), which is frequently used for inference that requires exchangeability assumptions, but no other distribution assumptions. Like classical confidence intervals, our confidence intervals are not valid in a distribution-free setting. Our assumptions are neither strictly stronger nor weaker than distribution-free approaches: we do not require the exchangeability assumption (because the spatial locations are fixed for us), but we do assume a parametric (Gaussian) model for the noise. We see in our experiments that approaches that are known to be valid in the distribution-free setting, like the sandwich estimator, do not provide correct coverage in the setting we study.
>
> **Extensions beyond Gaussian noise.** An asymptotic analysis of our method under weaker noise settings (e.g. bounded variance) would be an interesting area for further work. If one allows the number of neighbors to tend to infinity, then asymptotic normality of our estimator follows by the classical central limit theorem together with the continuous mapping theorem. However, a detailed and complete asymptotic study of our method would require significant additional mathematical development.
>
> **Restrictiveness of Assumption 1.** The reviewer suggests that Assumption 1 requires that, for a fixed spatial location, the covariate will be always the same. In fact, we do not need to assume that covariates are fixed across time in our framework. In particular, as we detail next, there are at least three ways that time can be accounted for within our framework, and which way is most appropriate depends on the scientific question of interest:
> 1. Time can be treated as another “spatial” dimension. The test points and train points both then have a temporal location as well as a spatial location. We might take this approach if we were interested in the association between hourly air temperature and hourly air pollution, including information for all hours from 2010–2020. In this case, we would need to assume that hourly average air pollution satisfies a Lipschitz assumption in both space and time.
> 2. For each spatial location, each covariate and response could be aggregated over time (for example, via taking a mean or median at each spatial location, particularly if at many points in time the same spatial location is observed). The regression analysis can then be run on this temporally-aggregated data. We might take this approach if we were interested in the association between decadal average temperature and decadal average air pollution in major US cities, averaged over 2010–2020.
> 3. Data from a single point in time can be analyzed. In this case, many covariates (e.g. temperature) are essentially fixed functions of space, since time has been fixed. We might take this approach if we were interested in the association between current temperature and current air pollution in major US cities at a particular fixed time
> We think that each type of question above is of practical interest. We also agree with the reviewer that there exist questions of practical interest for which our method is not appropriate (e.g., analyzing demographic data with multiple individuals in a single household). We will add more discussion of the various ways to handle time within our framework, as well as limitations, in an updated manuscript.
>
> We hope these clarifications address your questions.  Thank you again for your constructive feedback!
>
> **References:**
>
> Giordano, Ray, Schmidt-Hieber. On the inability of Gaussian process regression to optimally learn compositional functions. NeurIPS 2022.
>
> Zhang, Yuan, Zhu. Byzantine-tolerant federated Gaussian process regression for streaming data. NeurIPS 2022.
>
> El Harzli, Grau, Valle-Perez, Louis. Double-Descent Curves in Neural Networks: A New Perspective Using Gaussian Processes. AAAI 2024.
>
> Jordan. "Bayesian nonparametric learning: Expressive priors for intelligent systems." Heuristics, probability and causality: A tribute to Judea Pearl, 2010.
>
> Cui, Loureiro, Krzakala, Zdeborova. Generalization Error Rates in Kernel Regression: The Crossover from the Noiseless to Noisy Regime. NeurIPS 2021.
>
> Cherian, Gibbs, Candes. Large Language Model Validity via Enhanced Conformal Prediction methods. NeurIPS 2024.
>
> Correia, Massoli, Louizos,  Behboodi. An Information Theoretic Perspective on Conformal Prediction. NeurIPS 2024.

---

> > ### Comment · Reviewer_pd1Y · 2025-08-01
> > **Update**
> >
> > I thank the authors for the detailed feedback.
> > To my first question: as someone who got exposed to this problem for the first time, it would be beneficial for the authors to add summary sentences at the beginning of the paragraph so I know what to anticipate. Additionally, I think explicitly spelling out how models can be applied in this problem setting is also important before the authors introduce nonlinear approaches.
> >
> > To my second point: it never hurts to clarify what assumptions you need to avoid confusion with the use of the term nonparametric from other fields.
> >
> > I understand that the analysis of the problem that the authors are proposing might be nontrivial. However, the authors should explain the technical challenges in proving their main theorem and provide a proof sketch, as the main proof with the most general setting is included in the appendix. As linear regression is a classical approach, highlighting the technical difficulty of proving the results might be important. For example, CIs have been proposed in semi-parametric estimation without spatial association. Having a related work section to review different CI construction and why they can't be applied in the problem setup that the authors are focusing on can largely clarify the contribution of this paper.
> >
> > Additionally, the authors take a lot of effort to stress that a finite sample is an important reason why linear regression is important. However, their confidence interval is only valid (or has valid coverage) asymptotically with large N when $\sigma$ is unknown (which is often the case in practice). I understand this is standard in linear regression analysis, but some explanation would be needed to clarify this in the abstract and introduction.

---

> > > ### Author Response · Authors · 2025-08-04
> > > **Reply to reviewer's update**
> > >
> > > We thank the reviewer for taking the time to read and consider our response, and for the detailed feedback that we believe will lead to improvements in the clarity of our paper. We respond to each of their points in more detail next.
> > >
> > > **Clarity in introduction.** We are happy to add additional structure to the introduction to help readers understand where each paragraph is going in terms of motivating the problem. We thank the reviewer for this suggestion.
> > >
> > > **Use of the term nonparametric and Gaussian noise** We also will clarify our use of the term *nonparametric*, and ensure that we are clear that our analysis assumes Gaussian noise.
> > >
> > > **Related work and technical contribution** We included an extended related work in the appendix (due to space considerations). We are happy to add additional comparisons between our approach and existing semi-parametric approaches here. We will also add a short summary of this comparison, highlighting our technical contribution, in the main text.
> > >
> > > **Finite sample validity** We agree with the reviewer that we should clarify early in the text that precise finite-sample validity only holds when the noise variance is known. We will acknowledge this limitation in our theory in the paper when claiming finite-sample validity. We will also report estimated values for the noise variance in simulations, where we know the ground truth. In particular, our simulation results suggest that we obtain reasonable estimates of the noise variance. We believe that this empirical evidence, together with our asymptotic theory, further supports the claim that in practice we still expect our confidence intervals to be (roughly) valid in finite samples.

---

> > > > ### Comment · Reviewer_pd1Y · 2025-08-04
> > > > **updated text request**
> > > >
> > > > Could the authors please include the text that you plan to update here? I am happy to increase my score if they are clear.

---

> ### Author Response · Authors · 2025-08-08
> **Proposed changes to the text (1/2)**
>
> We thank the reviewer for being so engaged during the discussion period and for the opportunity to clarify our plans.
>
> **Clarity in introduction.** At the start of the paragraph at line 27, we will add "\paragraph{Estimator.} Our goal in the present work is to provide valid confidence intervals for an estimator of these associations. Our confidence intervals will be useful insofar as the estimator is useful. Therefore, we first argue that, despite many recent advances in machine learning, the simple estimator we focus on in the present work remains the most natural choice. Subsequently we describe the challenge in constructing valid confidence intervals.” And at line 62 we will emphasize the transition by starting with “\paragraph{Valid confidence intervals.}”
>
> **Use of the term nonparametric and Gaussian noise.** In lines 77--80, we will update the text with the following sentence: “Our principal contribution is to introduce the first method for constructing confidence intervals in spatial linear regression that guarantees frequentist coverage at the nominal level (a) under nonparametric assumptions ***on the form of the expected response as a function of space*** and (b) when the target locations differ from the observed ones.” In particular, we propose to replace the current phrase “data-generating distribution” with the words in bold italic.
>
> **Comparison to Semiparametric Inference for Partially Linear Model: Appendix.** We first propose the following text for the appendix. Our proposed main-text summary of this information then appears below.
>
> `start quote`
>
> Partially linear models take the form
> $$
> \mathbb{E}[Y |X, S|= \beta^T X + g(S)
> \mathbb{E}[X|S] = \chi(S),
> $$
> where $\beta$ is the parameter of interest, $S$ is a nuisance (or control) variable, $X$ is the covariates, and $g$ is an unknown and possibly complicated function. These models are widely studied in the semiparametric literature. Among many others, Robinson et al 1988, Robins et al 1992, Chernozukov et al 2018 focus on estimation of $\beta$, under the assumption that the triples $(S_n, X_n, Y_n)$ are independent and identically distributed across data indices $n$.
>
> In many spatial applications, it isn’t reasonable to think of the nuisance variable (geographic space) as sampled independently and identically, or even at regularly spaced locations. Observational data are often collected in a highly non-uniform way — densely in some regions, sparsely or not at all in others — due to physical constraints, accessibility, or policy decisions. This non-uniformity introduces distribution shifts when attempting to generalize inferences from one region to another. In the present work, we focus on inference with fixed spatial locations and do not impose regularity conditions on the sampling design. This setup allows us to quantify uncertainty in associations in cases where extrapolation to poorly-sampled geographic areas is required, or in cases with heavily clustered training locations and more uniform target locations.
>
> A notable exception to the assumption of fully i.i.d. data in the semiparametric literature is Heckman1986. Heckman 1986 considered time as a nuisance variable, and assumed this nuisance variable was one-dimensional and sampled densely and in a sufficiently regular way. In contrast, we allow for multiple spatial dimensions and do not require regularity assumptions about the sampling design.
>
> `end quote`

---

> > ### Author Response · Authors · 2025-08-08
> > **Proposed changes to the text (2/2)**
> >
> > **Comparison to Semiparametric Inference: Main Text Summary.** Since the distinction with our work is easiest to describe with some notation and an understanding of our method, we propose to put the following summary after our current Section 3 in a new main-text (additional) related work section. “Among many others, Robinson 1988, Robins et al 1992, and Chernozukov et al 2018 consider semiparametric inference in partially linear models. Partially linear models are linear in the covariates conditional on some ‘nuisance parameter’ (in our case, spatial location) but allow for a flexible relationship between the ‘nuisance parameter’ and the response, as well as a flexible relationship between nuisance parameters and the covariates. Robinson 1988, Robins et al 1992, and Chernozukov et al 2018 all consider triples of nuisance, covariate, and response variables (here, $(S_n, X_n, Y_n)$) — and assume these triples are i.i.d. across data indices $n$. In many spatial applications, it isn’t reasonable to think of the nuisance variable (geographic space) as sampled independently and identically, or even at regularly spaced locations. For instance, observational data are often collected in a highly non-uniform way — densely in some regions, sparsely or not at all in others — due to physical constraints, accessibility, or policy decisions. Since we instead provide valid confidence intervals for arbitrary, fixed spatial locations, we are able to quantify uncertainty in cases where spatial locations are observed through a nonrandom process and we want to make inference about a different set of locations.”
> >
> > **Finite Sample Validity.** We will clarify in both the abstract and introduction that our confidence intervals are valid in finite samples under homoskedastic Gaussian noise with known variance, and asymptotically valid if the variance is unknown. In the *Abstract (lines 11--12)* we will update the existing text to “Our method requires minimal assumptions beyond a form of spatial smoothness ***and a homoskedastic Gaussian error assumption***. In particular, we do not require model correctness or covariate overlap between training and target locations. Our approach is the first to guarantee nominal coverage in this setting and outperforms existing techniques in both real and simulated experiments. ***Our confidence intervals are valid in finite samples when the noise of the Gaussian error is known, and we provide an asymptotically consistent estimation procedure for this noise variance.***”  In the *Introduction (line 84)* we will update the text to “we instead assume the response is a smooth function of space ***observed with homoskedastic Gaussian noise. When the variance of this noise is known, our confidence intervals are valid in finite samples. To address the common case where the variance is unknown, we provide an asymptotically consistent estimator for it.***”
> >
> >
> > **References.**
> >
> > Robinson. Root-n-consistent semiparametric regression. Econometrica. 1988.
> >
> > Robins, Mark and Newey. Estimating exposure effects by modelling the expectation of exposure conditional on confounders. Biometrics. 1992.
> >
> > Chernozhukov, Chetverikov, Demirer, Duo, Hansen, Newey. Double/debiased machine learning for treatment and structural parameters. The Econometrics Journal. 2018.
> >
> > Heckman. Spline smoothing in a partly linear model. Journal of the Royal Statistical Society. Series B (Methodological). 1986.

---

> > > ### Comment · Reviewer_pd1Y · 2025-08-08
> > > **update**
> > >
> > > I thank the authors for their response. The added text has largely clarified my questions, and I have updated my score accordingly.

---

### Official Review · Reviewer_W9eu · 2025-07-10

**Clarity:** 4
**Significance:** 4
**Originality:** 4
**Rating:** 5
**Confidence:** 2

**Summary:**

The paper considers constructing confidence intervals for a linear regression problem under covariate shift, where both training and test data is associated with a "spatial information".

Concretely, with $\mathcal{S}$ being some metric space (which should be thought of as low-dimensional - e.g. earth surface), the authors consider a model where there is an (unkown) Lipschitz function $\chi : \mathcal{S} \to \mathbb{R}^D$, mapping the spatial location into covariates, and the (noisy) observations are given as $Y_i = f(S_i) + \varepsilon$ for some Lipschitz function $f: \mathcal{S} \to \mathbb{R}$. The algorithm observes $N$ tuples $(S_i, X_i, Y_i)$ for $S_i \sim \mathcal{D}_1$, such that $X_i = \chi(S_i)$, followed by $M$ tuples $(S_i\^*, X\_i\^\*)$ where $S_i$ is drawn from a potentially different distribution $\mathcal{D}_2$.

The relation between $Y_i$ and $X_i$ is not assumed to be linear -- the authors nevertheless suggest fitting a linear model in this misspecified case, attempting to approximate $Y_i$ as $\langle \theta, X_i\rangle$ as well as possible - as they claim even if it does not necessairly yield the best possible predictions for $Y^*$, it is more interpretable than alternatives, and often might provide qualitative understanding of which covariates $X_i$ have impact on $Y$.

As such, they attempt to provide a confidence intervals for coordinates of the least square linear regression parameter at the "target locations" $S_i^\*$
$\theta_{OLS} := \mathrm{arg min}\_{\theta} \sum\_{i \leq M} \mathbb{E} \left[(Y\_i^\* - \theta^T X\_i^\*)^2 | S\_i^\*\right],$
where, again $Y_i^\*$ are not observed by the algorithm, (and averaged over in the formula above)

To this end, they approximate $f(S\_i^\*) = \mathbb{E}[Y\_i^\* | S\_i^\*]$ as a weighted average $Y\_i$ for nearby points $S\_i$ in the training set (for experiments, they used simply putting weight $1$ on the nearest neighbor and $0$ on all remaining entries), producing estimated vector of responses $\hat{Y}\_i^\*$. They use this to compute $\hat{\theta}\_{OLS} = (X^{\*T} X^\*)^{-1} X^{\*T} \hat{Y}^\*$, and calculate the confidence interval (valid under the relatively weak assumption that the response $f(S)$ is Lipschitz in $S$) for coordinates $\hat{\theta}\_{OLS, p}$, by bounding bias introduced by approximating $Y\_i^\*$ with $\hat{Y}\_i^\*$. Thanks to the fact that the $\hat{Y}\_i^\*$ was taken as weighted averages of nearby points, obtaining bound on a bias reduces to a specific Wasserstein distance calculation.

**Questions:**

See "Strengths and weaknesses"; Are there some guidelines for choosing the Lipschitz constant for $f$? How likely is it in practice that the space is covered densely enough by the training data to provide useful confidence intervals? What is the effect on the mechanism for the choice of weights for the approximation of $\mathbb{E}[Y^* | S^*]$, on the results (and especially, how it depends on the density of the sampled points $S_i$ in the space $\mathcal{S}$?)

**Ethical Concerns:**

["NO or VERY MINOR ethics concerns only"]

**Final Justification:**

After clarification, I continue to recommend acceptance.

**Limitations:**

Some limitations have been addressed.

**Paper Formatting Concerns:**

No concerns.

**Quality:**

4

**Strengths And Weaknesses:**

The paper is extremely well motivated, with both natural and relevant setting - authors explicitly provide a number of applications across various sciences for which the data is associated with spatial information, and the distributions of the spatial variable differs on the training and test data. The linear regression is already ubiquitous in sciences, in particular due to being more interpretable than more complex models. As such, having reliable confidence interval for regression coefficients in the misspecified case is particularly helpful.

Assumption that the covariates are just a (deterministic) Lipschitz function of the spatial variable is a relatively strong limitation, but it has been explicitly acknowledged by authors - this work is a step in the right direction, and as they mention, there is a room for further research. It seems reasonable to assume, on the other hand, that in many scenarios the conditional expectation of the response variable on the spatial variable is Lipschitz in the spatial variable.

It is unclear to me, how to pick a specific Lipschitz constant for the dependence of the expected response on the spatial variable, in a given practical situation, and moreover - how often one should expect that the space $\mathcal{S}$ is covered by a training data within the radius much smaller than the chosen Lipschitz constant - this already shows up in Figure 3 (the comparison of this method with other methods for the tree cover experiment): while the proposed method, is the only one of the compared ones that achieves a nominal coverage (i.e. confidence intervals are reliable), the width of the confidence intervals is an order of magnitude larger than the one produced by, for example, GP-BCI, and the coverage is in fact way larger than one would intuitively deem necessary - which might lead to unnecesairly rejecting many potential scientific results as statistically insignificant - arguably, the problem less prevalent than the opposite problem of false-positives in science.

The proposed framework allows for approximating $\mathbb{E}[Y^\* | S^\*]$ as a weighted linear combination of nearby training points, but in most calculations a very simple weight distribution is used, specifically the "nearest neighbor" weight $1$ put on the nearest neighbor of $S^\*$ in the dataset. Slightly more discussion on why this choice was made, and how it affects the results, particularly depending on how densely space $\mathcal{S}$ is covered by the training data. This seems like a somewhat unnatural choice, as it approximates a Lipschitz function $f$ by essentially piecewise constant function on the Voroni diagram given by the data points $S_i$ -- which is far from Lipschitz. The authors briefly mention $K$-nearest neighbors as a different choice; another natural choice would be to use weights proportional to, say, gaussian kernel i. e. $w\_i \propto \exp(-d(S^\*, S\_i)^2 / L^2)$; it is slightly more discussion (and potentially a follow-up work) on how this choice affects the results would be great.

---

> ### Author Rebuttal · Authors · 2025-07-30
>
> We are grateful to the reviewer for their careful reading, insightful comments, and detailed, thoughtful feedback. We are excited to hear that the reviewer found our paper well-motivated and our setting natural and relevant.
>
> **Assumptions on the covariates:** (*“Assumption that the covariates are just a (deterministic) Lipschitz function of the spatial variable is a relatively strong limitation”*)
>
> We see two potential assumptions that may be of concern here: (A) assuming the covariates are Lipschitz in the spatial variables, and (B) assuming that the covariates are a deterministic function of the spatial variables. To address (A), we want to start by clarifying that we do not require a Lipschitz assumption on the covariates as a function of the spatial variables. The (only) smoothness assumption we make (namely, Assumption 4) is on the conditional expectation of the response given the spatial variables. For point (B), we agree with the reviewer that requiring the covariates to be a deterministic function of space (namely, Assumption 1) means our method may not apply straightforwardly to certain types of data (e.g., demographic data with multiple individuals in a single household). However, we think there remain many practical settings of interest where Assumption 1 is not problematic; for instance, consider covariates such as precipitation, humidity, temperature, or particulate matter concentration — and either treat time as a spatial variable, average across a particular time range of interest, or consider a fixed time point.
>
> **Choice of Lipschitz constant:** Choosing the Lipschitz constant requires user knowledge about the specific domain of the problem. Generally, especially in scenarios requiring extrapolation, the Lipschitz constant cannot be reliably estimated solely from the observed data without additional assumptions — since function values are not available far from observed locations. We emphasize that we selected the Lipschitz assumption specifically because it is relatively interpretable in physical and spatial contexts compared to common statistical assumptions (e.g., that the function lies within the unit ball of a reproducing kernel Hilbert space). We believe that some form of smoothness assumption is often made, implicitly or explicitly, when working with spatial data. And making this assumption explicit is preferable to leaving it implicit.
>
> For a concrete example, consider the problem of selecting a Lipschitz constant in an analysis where the response is annual average PM2.5 over California. Chow et al. 2006 claim that “Zones of representation for PM2.5 varied from 5 to 10 km for the urban Fresno and Bakersfield sites, and increased to 15–20 km for the boundary and rural sites” where “[t]he zone of representation is defined as the radius of a circular area in which a species concentration varies by no more than ±20% as it extends outward from the center monitoring site”. The annual PM2.5 concentrations in the study area do not exceed 30 ug/m^3. The combination of a zone of representation between 5 and 20 km, and a variation of not more than 30 ug/m^3 within this zone of representation suggests a range of possible Lipschitz constants: 0.25–1.2 (ug/m^3) / km. The authors also point out that topographical and meteorological phenomena contribute to this scale of variation. So we would not expect this proposed constant to be “universal” for problems related to PM2.5, but we might expect that this range of Lipschitz constants is a reasonable starting point for other studies involving annual average PM2.5 with similar weather and topography to California. We showed in our real-data analysis that a range of Lipschitz constants can still produce qualitatively similar results (Figure 9) and correct coverage. To err towards the side of a conservative analysis, we would recommend a user selects the largest Lipschitz constant in this range (i.e. 1.2 (ug/m^3)/km).
>
> **Width of confidence intervals:** We appreciate the reviewer’s point regarding the substantial width of confidence intervals produced by our method in certain experiments, such as the tree cover experiment (Figure 3). We agree that excessively wide intervals could limit practical utility and potentially lead to overly conservative conclusions, rejecting otherwise meaningful scientific findings as insignificant. We highlight that the increased width arises naturally from our explicit control of bias under extrapolation conditions, especially when the test locations significantly differ from training locations. The large interval widths observed in Figure 3 result directly from this extrapolation scenario, reflecting genuine uncertainty rather than methodological conservatism. We also want to point out that our experiments clearly illustrate that existing confidence intervals typically fail to achieve the nominal coverage — and often have near-zero coverage (as in Figs. 1 & 2). Thus, our method yields the narrowest intervals among all evaluated methods that consistently achieve nominal coverage. We emphasize that producing narrower intervals by sacrificing validity is trivial: a zero-width confidence interval achieves minimal width but does not yield meaningful coverage guarantees. Hence, the widths of invalid intervals are not directly comparable to those of valid intervals. We also note explicitly that our intervals are practically meaningful: in many cases shown in our experiments, intervals are narrow enough to exclude zero, providing clear scientific conclusions about the sign and magnitude of effects. We will explicitly discuss and illustrate these points in the revised manuscript.
>
> **Practical coverage and training data density:** The reviewer rightly raises a practical question regarding how densely the spatial domain must be covered by training data to yield useful confidence intervals. In practice, we expect that our intervals will be relatively narrow whenever the data spacing is substantially smaller than the scale implied by the Lipschitz constant (the distance scale over which meaningful response variations occur). Conversely, if sampling is sparse relative to the Lipschitz scale, intervals will naturally widen. Since this increasing width represents legitimate uncertainty due to extrapolation, we would argue that our intervals are still useful in this case, just (appropriately) wider than the dense-data case. We will explicitly discuss this point in the revised manuscript, clarifying when we can expect our method to provide narrow intervals and how this relates to spatial data density and the Lipschitz constant.
>
> **Effect of weighting mechanism choice:** The reviewer correctly points out that our method allows general weighting schemes for approximating the conditional expectation. Our current choice — nearest-neighbor weighting (weight = 1 for the closest point) — is intentionally simple, transparent, and yields the tightest possible Lipschitz bound on bias. It simplifies derivations and clearly demonstrates our theoretical guarantees. We agree that alternative weighting schemes (such as k-nearest neighbors or the Gaussian kernel suggested by the reviewer) would also be natural choices. These methods produce smoother approximations at the expense of slightly looser bias bounds, potentially reducing variance and practical interval widths. The benefit of smoother weighting schemes generally becomes more pronounced as the density of spatial sampling increases since smoother weighting schemes naturally leverage multiple nearby points, reducing variance while controlling bias effectively. We believe exploring this tradeoff is an interesting and practically relevant direction. We will include additional discussion and comparisons between these weighting schemes in the revised manuscript, explicitly highlighting how data density impacts these trade-offs. We also agree with the reviewer that follow-up work exploring these weighting schemes would be very valuable.
>
> We hope these clarifications address your questions.  Thank you again for your constructive feedback!
>
> **References:**
>
> Chow, Chen, Watson, Lowenthal, Magliano, Turkiewicz, Lehrman. PM2.5 Chemical Composition and Spatiotemporal Variation During the California Regional PM10/PM2.5 Air Quality Study. Journal of Geophysical Research. 2006.

---

### Decision · Program_Chairs · 2025-09-17

**Decision:**

Accept (poster)

**Comment:**

We thank the authors for their submission.

The authors propose a confidence interval estimator in a spatial regression setting when a linear model may be misspecified.  They show that given a smoothness assumption on the response, their approach can achieve nominal coverage. The method is evaluated on synthetic and real data, favorably comparing to alternative approaches.

Reviews were uniformly positive.  Reviewers agreed that this work is well motivated, well-organized and (mostly) clearly written, and technically solid.  The authors clearly state the limitations of their approach (i.e., the strength of the smoothness assumption).  The discussion with reviewer W9eu shed light on the practical implications of this limitation, with the remedy to incorporate domain knowledge about the response function.  The discussion with reviewer pd1Y led to multiple clarifications in the text, as well as additional discussion about the specific technical contribution of this work and the existing literature.